

# Implementing nonlinear viscoplasticity in ASPECT: benchmarking and applications to 3D subduction modeling

Anne Glerum[1], Cedric Thieulot[1], Menno Fraters[1], Constantijn Blom[1], and Wim Spakman[1,2]

[1]Earth Sciences, Utrecht University, Utrecht, Netherlands
[2]Centre of Earth Evolution and Dynamics (CEED), University of Oslo, 0316 Oslo, Norway

*Correspondence to:* Anne Glerum (A.C.Glerum@uu.nl)

**Abstract.** ASPECT (Advanced Solver for Problems in Earth's ConvecTion) is a massively parallel finite element code originally designed for modeling thermal convection in the mantle with a Newtonian rheology. The code is characterized by modern numerical methods, high-performance parallelism and extensibility. This last characteristic is illustrated in this work: we have extended the use of ASPECT from global thermal convection modeling to upper mantle-scale applications of subduction.

Subduction modeling generally requires the tracking of multiple materials with different properties and with nonlinear viscous and viscoplastic rheologies. To this end, we implemented a frictional plasticity criterion that is combined with a viscous diffusion and dislocation creep rheology. Because ASPECT uses compositional fields to represent different materials (and, since recently, tracers), all material parameters are made dependent on a user-specified number of fields.

The goal of this paper is primarily to describe and validate our implementations of complex, multi-material rheology by reproducing the results of four well-known two-dimensional benchmarks: the indentor benchmark, the brick experiment, the sandbox experiment and the slab detachment benchmark. Furthermore, we aim to provide hands-on examples for prospective users by demonstrating the use of multi-material viscoplasticity with three-dimensional, thermomechanical models of oceanic subduction, putting ASPECT on the map as a community code for high-resolution, nonlinear rheology subduction modeling.

## 1 Introduction

Earth is a complex dynamic system that deforms on a wide range of spatial and temporal scales. To obtain realistic predictions of this system from numerical simulations, it is key to capture the relevant aspects of this deformation behavior. Here we are concerned with the longer geological timescales of the subduction of lithospheric plates into the mantle. On such timescales, rock deformation is mostly non-elastic and characterized by unrecoverable solid-state creep and brittle-plastic failure (Ranalli, 1995; Karato, 2008; Burov, 2011). Strain rate dependent viscous deformation through the mechanism of solid–state creep is dominated by linear (Newtonian) diffusion creep and various forms of nonlinear high and low temperature dislocation creep (e.g. Ranalli, 1995; Burov, 2011). Plastic yielding occurs when large differential stresses cause rocks to fail beyond the creep regime by local brittle fracture or, at higher temperatures, through ductile homogeneous material flow (Ranalli, 1995; Karato, 2008).



The implementation of plastic yielding into numerical modeling software entails the definition of a yield criterion that the maximum stress must satisfy (Davis and Selvadurai, 2002). Several different plastic yield criteria, such as the Mohr–Coulomb, Drucker–Prager or the Griffith–Murrell criteria (see Braun, 1994; Braun and Beaumont, 1995; Davis and Selvadurai, 2002; Kachanov, 2004, and references therein), are commonly used. These formulations introduce a pressure dependence (frictional plasticity) in the yield criterion. Whereas failure behavior is similar between different rock types and depends primarily on pressure (Burov, 2011), deformation in the viscous creep regime (when stresses are below the plastic yield strength) requires the implementation of rheological descriptions varying with rock type, pressure, temperature, strain rate and other factors such as grain size and water content (Burov, 2011). The implementation of plastic failure and viscous creep complicates solving the governing equations of flow problems due to the nonlinear dependence of the so-called effective viscosity on solution variables strain rate, pressure and temperature (Gerya, 2010). However, the necessity of using viscoplastic rheologies for simulating natural deformation processes, particularly of the lithosphere, is generally accepted.

Meanwhile many 3D geodynamical codes offer modeling using complex nonlinear viscoplastic rheology, examples of such advanced codes are (in alphabetical order) CitcomCU (Moresi et al., 1996; Zhong, 2006), DOUAR (Braun et al., 2008), FANTOM (Thieulot, 2011), Fluidity (Davies et al., 2011), I3(E)LVIS (Gerya and Yuen, 2007), LaMem (Binder et al., 2016), MILAMIN (Dabrowski et al., 2008), pTaTin3D (May et al., 2015), Rhea (Burstedde et al., 2008), Slim3D (Popov and Sobolev, 2008), TERRA (Baumgardner, 1985; Davies et al., 2013) and Underworld2 (Moresi et al., 2007).

To this list we can now add the recent open source code ASPECT (Advanced Solver for Problems in Earth's ConvecTion; Kronbichler et al. (2012)), which was originally designed for modeling thermal convection in the mantle. ASPECT is a massively parallel finite element code that is based on state-of-the-art numerical methods, such as high-performance iterative and direct solvers and adaptive mesh refinement, to solve problems of both compressible and incompressible flow. It builds on tried-and-well-tested libraries such as deal.II (Bangerth et al., 2007, 2012), Trilinos (Heroux et al., 2005, 2014) and p4est (Burstedde et al., 2011) and is under constant development (Bangerth et al., 2016; ASPECT, 2016; Dannberg and Heister, 2016; Rose et al., 2017).

However, ASPECT originally did not include modeling with multiple nonlinear viscoplastic materials as needed for, e.g., longterm tectonics modeling. Therefore we implemented and benchmarked a frictional plasticity (Drucker–Prager) criterion that can be combined with a viscous creep rheology (diffusion, dislocation or composite creep) for any number of materials, allowing for fully thermomechanically coupled viscoplastic flow, on which we here report. There are two papers that use AS-PECT that employ a simpler, one-material viscoplastic rheology for planetary convection (Tosi et al., 2015; Zhang and O'Neill, 2016). Here we focus on benchmarking our implementations in light of lithospheric deformation and these implementations as well as example model set-ups have or will become part of ASPECT, together with extensive documentation, providing hands-on applications of the code. We show that our viscoplastic rheology description enables extending applications beyond thermal mantle convection to detailed lithospheric subduction modeling.

We first present the algorithms underpinning the ASPECT code and our additions pertaining to rheology and compositional fields (Section 2). We then validate our implementations in Section 3 using four benchmarks of increasing complexity: the indentor benchmark (Thieulot et al., 2008; Thieulot, 2014), the brick experiment (Lemiale et al., 2008; Kaus, 2010), the



numerical sandbox (Buiter et al., 2006) and the slab detachment benchmark (Schmalholz, 2011; Thieulot et al., in prep.). Finally, in Section 4 we present two 3D subduction applications to showcase the new suite of possibilities made available through our additions and adaptations, and we discuss our overall results in Section 5.

## 2 Methods

A short summary of the governing equations solved by ASPECT is given in Section 2.1 (for more information the reader is referred to Kronbichler et al., 2012). Section 2.2 lists our specific additions to the code.

### 2.1 ASPECT

#### 2.1.1 Governing equations

ASPECT can solve for both compressible and incompressible flow, but here we focus on the latter, adopting the Boussinesq approximation and assuming an infinite Prandtl number (i.e. inertial term is omitted). Heat production is not incorporated. This results in the following equations of conservation of momentum (1), mass (2) and energy (3):

$$-\nabla \cdot (2\mu\dot{\epsilon}(\boldsymbol{u})) + \nabla P = \rho\boldsymbol{g} \tag{1}$$

$$\nabla \cdot \boldsymbol{u} = 0 \tag{2}$$

$$\frac{\partial T}{\partial t} + \boldsymbol{u} \cdot \nabla T - \nabla \cdot (\kappa + \nu_h(T)) \nabla T = 0 \tag{3}$$

where density $\rho = \rho_0(1 - \alpha(T - T_0))$. Other symbols are explained in Table 1. Artificial diffusivity $\nu_h$ is used to prevent oscillations due to the advection of the temperature field. It is calculated according to the entropy viscosity method of Guermond et al. (2011), as described in Kronbichler et al. (2012).

Similar to the description of temperature, distinct sets of material parameters are represented by *compositional fields* that are advected with the flow. For each field $c_i$, this formulation introduces an additional advection equation (4) to the system of equations (1)–(3) described above. As these equations contain no natural diffusion, artificial diffusivity $\nu_h$ is again introduced to stabilise advection:

$$\frac{\partial c_i}{\partial t} + \boldsymbol{u} \cdot \nabla c_i - \nabla \cdot (\nu_h(c_i)) \nabla c_i = 0 \tag{4}$$

Note that as of 2016 it is also possible to use active as well as passive tracers in ASPECT (version 1.4.0).

#### 2.1.2 Solving the governing equations

ASPECT solves the above equations using the finite element method: the domain is discretized into quadrilateral/hexahedral finite elements and the solution (velocity, pressure, temperature and compositional fields) is expanded using Lagrange polynomials as interpolating basis functions. Default settings employ second order polynomials for velocity, temperature and composition and first order polynomials for pressure ($Q_2Q_1$ elements, e.g. Donea and Huerta, 2003). Unless stated otherwise,





these default polynomial degrees are used in the following. The linearized Stokes system is solved in a procedure involving the iterative FGMRES solver with an inexact right preconditioner. For details on the construction of the preconditioner, see Kronbichler et al. (2012). The CG method with an incomplete LU decomposition preconditioner is used for the temperature and composition systems. Nonlinearities in the rheology are resolved with Picard-type (fixed point) iterations, iteratively up-

dating the velocity and pressure, strain rate and viscosity (Ismail-Zadeh and Tackley, 2010) until the relative nonlinear residual $\frac{||\mathbf{A}(\boldsymbol{x}_i)\boldsymbol{x}_i - \boldsymbol{b}||_2}{||\mathbf{A}(\boldsymbol{x}_0)\boldsymbol{x}_0 - \boldsymbol{b}||_2}$ has fallen below a user-set tolerance (default value of $10^{-6}$). The initial residual $||\mathbf{A}(\boldsymbol{x}_0)\boldsymbol{x}_0 - \boldsymbol{b}||_2$ is computed with zero velocities.

## 2.2 Additions to ASPECT

### 2.2.1 Nonlinear rheologies

The ASPECT code is divided into different modules for boundary conditions, initial conditions, mesh refinement etc. Each module comprises of several plug-ins providing different implementations (e.g. constant vs. space and time dependent boundary conditions), to which the user can add its own if more functionality is needed. Rheologies are implemented within the so-called *Material model* module. Plug-ins in this module must provide functions that compute the viscosity, density, thermal conductivity, thermal diffusivity, specific heat and the thermal expansion coefficient at the quadrature points. The solution variables

$\dot{\epsilon}(\boldsymbol{u})$, $T$, $P$ and $c_i$ as well as position are available to compute these material properties. This then provides a straightforward way of implementing nonlinear rheologies, which we have taken advantage of.

Deformation of materials at longer timescales is predominantly defined by brittle fracture or viscous creep in terms of diffusion and dislocation creep at relatively low stresses (Karato, 2008). We thus implement three basis rheologies that can be combined into more complex ones:

1. Grain boundary or bulk diffusion creep

2. Power-law dislocation creep

3. Plastic yielding

Rheologies 1 and 2 can be conveniently formulated with one equation (Karato and Wu, 1993; Karato, 2008):

$$\mu_{eff}^{vsc} = \frac{1}{2} K \left(\frac{d}{b}\right)^{m/n} \left(\frac{1}{A}\right)^{1/n} \dot{\epsilon}_e^{(1-n)/n} \exp\left(\frac{Q+PV}{nRT}\right) \tag{5}$$

where in case of diffusion creep, $n=1$ and $m>0$, while for dislocation creep $n>1$ and $m=0$. See Table 1 for the definition of used symbols. The effective deviatoric strain rate is defined as $\dot{\epsilon}_e = \sqrt{\frac{1}{2}\dot{\epsilon}_{ij}'\dot{\epsilon}_{ij}'}$. In this paper we report model set-up values for simplified prefactor $B$, defined as $\left(\frac{1}{B}\right)^{\frac{1}{n}} = \left(\frac{d^m K^n}{A b^m}\right)^{\frac{1}{n}}$, and add a scaling factor $\beta$ to Eq. (5) to easily tune the effective viscosity:

$$\mu_{eff}^{vsc} = \frac{1}{2} \beta \left(\frac{1}{B}\right)^{1/n} \dot{\epsilon}_e^{(1-n)/n} \exp\left(\frac{Q+PV}{nRT}\right) \tag{6}$$





Plastic yielding (rheology 3) is implemented by locally rescaling the effective viscosity in such a way that the stress does not exceed the yield stress, also known as the Viscosity Rescaling Method (VRM) (Willett, 1992; Kachanov, 2004). The effective plastic viscosity is thus given by

$$\mu_{eff}^{pl} = \frac{\sigma_y}{2\dot{\epsilon}_e} \tag{7}$$

where $\sigma_y$ is the yield value. In our implementation it is defined by the Drucker–Prager criterion (Davis and Selvadurai, 2002):

$$\sigma_y = C\cos(\phi) + \sin(\phi)P \qquad (2D) \tag{8}$$

$$\sigma_y = \frac{6C\cos(\phi)}{\sqrt{3}(3+\sin(\phi))} + \frac{6\sin(\phi)P}{\sqrt{3}(3+\sin(\phi))} \qquad (3D) \tag{9}$$

where dilatancy is neglected for simplicity. In case internal friction angle $\phi$ is zero, this criterion reverts back to the von Mises criterion.

Both types of viscous creep act simultaneously (Karato, 2008) under the same deviatoric stress, so their contributions to the effective viscosity are harmonically averaged into a composite viscosity (van den Berg et al., 1993):

$$\mu_{eff}^{cp} = \left(\frac{1}{\mu_{eff}^{df}} + \frac{1}{\mu_{eff}^{dl}}\right)^{-1} \tag{10}$$

To combine plastic yielding and viscous creep, we assume they are independent (parallel) processes (Karato, 2008), i.e. the mechanism resulting in the lowest effective viscoplastic viscosity is favored:

$$\mu_{eff}^{vp} = \min\left(\mu_{eff}^{cp}, \mu_{eff}^{pl}\right) \tag{11}$$

However, for a smoother transition between the different deformation regimes (which should be easier for the numerical scheme to solve), we also experimented with a harmonic average (following Ismail-Zadeh and Tackley, 2010):

$$\mu_{eff}^{vp} = \left(\frac{1}{\mu_{eff}^{cp}} + \frac{1}{\mu_{eff}^{pl}}\right)^{-1} = \left(\frac{1}{\mu_{eff}^{df}} + \frac{1}{\mu_{eff}^{dl}} + \frac{1}{\mu_{eff}^{pl}}\right)^{-1} \tag{12}$$

Because of the strain rate dependence of viscosity and the lack of an initial guess for the strain rate for the first timestep, a
20 user-defined initial viscosity $\mu_{init}$ is adopted for each compositional field, or an initial uniform strain rate $\dot{\epsilon}_{init}$ is set. We find that the values of $\mu_{init}$ and $\dot{\epsilon}_{init}$ can significantly affect the compute time of the first time step. During subsequent timesteps, the strain rate of the previous timestep is used as an initial guess for the iterative process.

The final effective viscosity $\mu_{eff}^{vp}$ is capped by the user-defined minimum viscosity $\mu_{min}$ and maximum viscosity $\mu_{max}$ to avoid extreme excursions and to ensure stability of the numerical scheme:

$$\mu_{eff} = \min(\max(\eta_{eff}^{vp}, \eta_{min}), \eta_{max}) \quad \text{or} \tag{13}$$

$$\mu_{eff} = \mu_{min} + \left(\frac{1}{\mu_{max}} + \frac{1}{\mu_{eff}^{vp}}\right)^{-1} \tag{14}$$

We have successfully run the models presented here with $\mu_{max}/\mu_{min}$ up to 7 orders of magnitude. Such a range covers the mantle viscosity profiles suggested in most literature, for example as summarized in Cizkova et al. (2012), and we assume that viscosities higher than $\mu_{max}$ do not change the behavior significantly.



### 2.2.2 Multiple compositional fields

Lithospheric geodynamic models often require the specification of materials with different properties, for example a light and weak upper crust versus a denser and stronger lithospheric mantle. To provide the functionality needed for geodynamic modeling, all major material properties of our *Material model* plug-in depend on any number of fields, as defined by the user
5 (composition dependent parameters are denoted with an asterisk in Table 1).

The use of multiple compositional fields raises the question of how to average their properties (viscosity, density and other). We have implemented the four averaging schemes commonly referred to in the literature (e.g. Deubelbeiss and Kaus, 2008; Schmeling et al., 2008):

$$\overline{\mu} = \frac{\sum\limits_{i=1}^{nc} c_i}{\sum\limits_{i=1}^{nc} \frac{c_i}{\mu_i}} \qquad \text{(harmonic)} \qquad (15)$$

$$\overline{\mu} = 10^{\left(\frac{\sum\limits_{i=1}^{nc} c_i \log_{10}(\mu_i)}{\sum\limits_{i=1}^{nc} c_i}\right)} \qquad \text{(geometric)} \qquad (16)$$

$$\overline{\mu} = \frac{\sum\limits_{i=1}^{nc} c_i \mu_i}{\sum\limits_{i=1}^{nc} c_i} \qquad \text{(arithmetic)} \qquad (17)$$

$$\overline{\mu} = \mu_{\max\limits_{i=1,\dots,nc}(c_i)} \qquad \text{(infinite norm)} \qquad (18)$$

where $nc$ is the total number of compositional fields $c_i$ in the domain.
The above methods have been shown to affect model results in the context of subduction: Schmeling et al. (2008) showed that the subduction process can be up to three times faster between one averaging method and the other, and the effect of mesh resolution on subduction evolution varies per method as well. Unless stated otherwise, we use the infinite norm rule in this paper; for a discussion of this choice, see Appendix A.

### 3 Nonlinear rheology benchmarks

To test and validate our implementation of multi-material viscoplastic rheologies, we performed four 2D experiments: the indentor benchmark, the brick experiment, the numerical sandbox and the slab detachment experiment. The experiments increase in the number of materials and in the complexity of the rheology used, as outlined in Table 2. Consequently, each experiment highlights different parts of the implementation and the functionalities of ASPECT.

All experiments were conducted on an in-house computer with $\sim 1,000$ cores (2.34GHz, 2.88Tb RAM memory, Qlogic
InfiniBand). Wall times quoted can have changed with versions of ASPECT newer than those used for the described experiments.



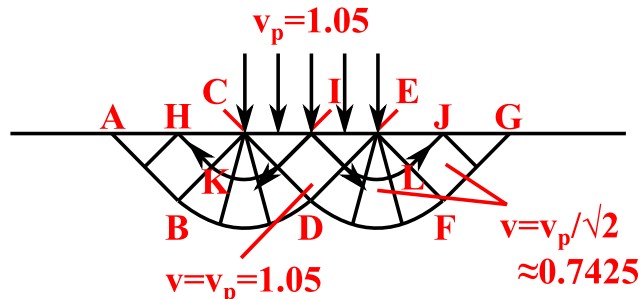

**Figure 1.** Prandtl's analytical solution of a rigid die indenting a rigid-plastic half space (Davis and Selvadurai, 2002; Kachanov, 2004; Thieulot et al., 2008) for a punch velocity $v_p$ of 1.05.

### 3.1 The indentor benchmark

In the indentor benchmark, a rigid indentor "punches" a rigid-plastic half space. The exact solution to this boundary value problem is given by slip line field theory (Davis and Selvadurai, 2002; Kachanov, 2004; Thieulot et al., 2008, Appendix). The analytical solution (Fig. 1) is characterized by 3 observations:

1. The angles of the shear bands stemming from the edges of the indented area are 45 degrees.

2. The pressure at the surface in the centre of the punch (I) and the pressure in triangles ABC & EFG is $P_I = \sigma_y(1+\pi)$ and $P_{ABC} = P_{EFG} = \sigma_y$, respectively.

3. The velocity magnitude in areas CDE and ABDC & EDFG is $v_{CDE} = v_p$ and $v_{ABDC} = v_{EDFG} = \frac{v_p}{\sqrt{2}}$, respectively.

### 3.1.1 Model set-up

The numerical set-up of the instantaneous indentor benchmark comprises a 2D unit square of purely plastic von Mises material, i.e. its yield value $\sigma_y$ is independent of pressure and remains constant. The material's upper boundary is punched along a distance $p$ by prescribing an inward vertical velocity $v_p$ on the otherwise open (stress free) boundary (see Fig. 2a). The horizontal component of velocity along $p$ is either set to zero or left free to implement the so called "rough" and "smooth" punch (Lliboutry, 1987; Lee et al., 2005; Thieulot et al., 2008), respectively, where the smooth punch assumes a frictionless contact
between the punched medium and the indentor. Model and numerical parameters of the performed indentor experiments are presented in Table 3.

### 3.1.2 Model results compared to the analytical solution

Figure 3 shows the model results for a rough (left column) and smooth (right column) punch. The obtained solutions agree with the analytical solution according to criteria 1–3 listed above. Outside the slip lines, viscosity is uniformly high. The low
viscosity shear bands fit the analytical slip lines well (Fig. 3a&e) and stem from the edges of the indentor at a 45 degrees angle





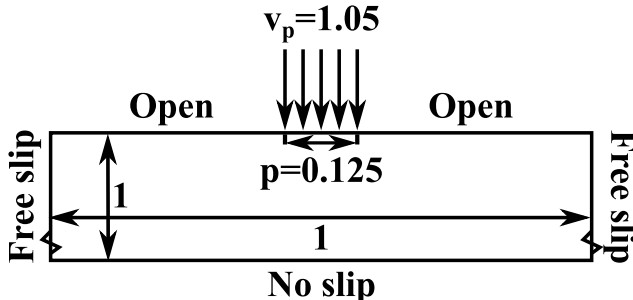

**Figure 2.** The indentor benchmark model set-up: a unit square with free slip vertical and no slip lower boundaries. The punch area has a prescribed vertical velocity $v_p$, the rest of the upper boundary is open.

with respect to the top of the medium (Fig. 3b&f). For a rough punch, pressure and velocity measurements deviate from the analytical solution by about 15% and 2%, respectively, but the block-like behavior of triangle CDE is evident (Fig. 3c). The analytical solution is exactly reproduced for a smooth punch (errors $< 0.2\%$), but the velocity vectors in Fig. 3g show some horizontal motion of triangle CDE and the velocity field is more diffuse. This trade-off is as expected, because the horizontal

component of surface velocity is left free for the smooth punch, while it is fixed to zero for the rough punch.

### 3.1.3 Discussion

ASPECT successfully reproduces the analytical solution of Prandtl for the rigid-plastic indentor benchmark, a problem with mixed boundary conditions and a nonlinear rigid-plastic rheology with overall viscosity contrasts of 6 orders of magnitude.

It should be noted that there exists a second end-member solution geometry for the smooth punch problem: Hill's solution

(Kachanov, 2004). Although Kachanov (2004) argues that Hill's solution is probably more correct when considering elasticity theory, and Lliboutry (1987) lists Hill's solution for the smooth punch, other numerical studies do not recover this slip line geometry in 2D either. In fact, our Prandtl shear band geometry compares well with results of Gerbault et al. (1998, Fig. 6b, smooth), Huh et al. (1999, Fig. 1, smooth), Christiansen and Pedersen (2001, Fig. 10, smooth), Zienkiewicz et al. (1995, Fig. 24–27, rough), Gourvenec et al. (2006, Fig. 6a, rough) and Yu and Tin-Loi (2006, Fig. 11, rough). In 3D, literature does suggest

that a rough interface between indentor and medium results in a Prandtl slip-line geometry, while Hill's solution is invoked by a smooth surface. Compare, for example, Fig. 11a and 11b of Gourvenec et al. (2006), Fig. 10e and 10f of Thieulot et al. (2008) or Fig. 13a and 13d of Braun et al. (2008).

The performed indentor experiment also shows a trade-off between accuracy in pressure and velocity measurements and the rigid-plastic like behaviour of the medium (compare left and right column of Fig. 3). This same dichotomy is seen however

in other studies that performed the experiment. Note, for example, the continuous velocity vectors in Huh et al. (1999). Also, Thieulot (2014) shows that the pressure under the punch improves from a 15% error to a mere 0.5% error when switching from a rough to a smooth footing.





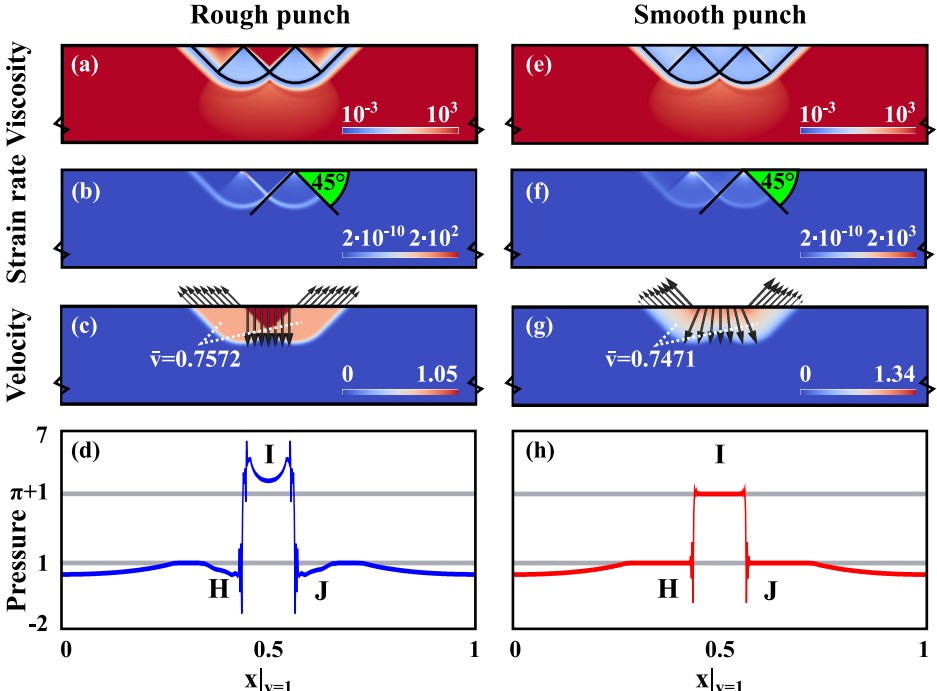

**Figure 3.** The punch benchmark results after 500 NI for a rough punch (left column) and a smooth punch (right column). (a) & (e): Viscosity field with analytical slip lines. (b) & (f): Strain rate field with measured shear band angles. (c) & (g): Velocity field with velocity vectors along the surface of the domain and velocity measurements in points K and L. (d) & (h): Pressure along the surface of the domain (colored line) and analytical solution values $\pi + 1$ and 1 (grey lines). Rough punch: $P_I = 4.7348$ and $P_H = P_J = 0.6983$. Smooth punch: $P_I = 4.1413$ and $P_H = P_J = 0.9997$.

## 3.2 The brick experiment

The brick benchmark has been used to investigate the numerical stability of shear band angles $\theta$ and their dependence on the internal angle of friction $\phi$ by Kaus (2010) and references therein. Three theoretical relationships have been proposed (Vermeer, 1990):

1. $\theta = 45 \pm \frac{\psi}{2}$     (Roscoe)

2. $\theta = 45 \pm \frac{\phi + \psi}{4}$   (Arthur)

3. $\theta = 45 \pm \frac{\phi}{2}$     (Coulomb)

where $\psi$ is the dilation angle (assumed to be zero in our case of incompressibility).





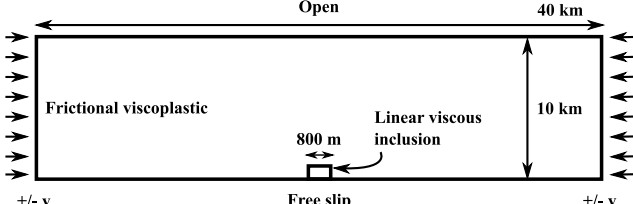

**Figure 4.** The brick benchmark model set-up after Kaus (2010): a rectangular domain with a prescribed inward or outward horizontal component of velocity on the vertical boundaries (the vertical component is left free). The upper boundary is open, while the bottom boundary is free slip. A small viscous inclusion of $800 \times 400$m is placed at the bottom of the domain.

### 3.2.1 Model set-up

In our instantaneous version of the brick benchmark, a viscous-frictional plastic medium with a small viscous inclusion at the bottom boundary (Fig. 4) is either compressed or extended (Lemiale et al., 2008; Kaus, 2010). Compression and extension are prescribed through kinematic boundary conditions on the vertical domain walls. The bottom boundary of the $40 \times 10$km domain is set to free slip and the top boundary is stress free (material is free to flow in or out). Other domain characteristics and material parameters are given in Table 4.

The angle of internal friction $\phi$ (Eq. 8) is varied from $0°$ to $30°$ to test the pressure dependency of the implemented plasticity criterion. It is expected that the resultant shear band angle $\theta$ varies with the internal friction angle. Shear band angles are automatically computed from the location of the maximum Frobenius norm of the strain rate $\sqrt{\dot{\epsilon} : \dot{\epsilon}}$ at $x = 17.4$km, $x = 19.4$km, $x = 20.6$km and $x = 22.6$km (Kaus, 2010).

### 3.2.2 Model results compared to the theoretical solution

Figure 5 depicts the measured shear band angle versus the supplied internal friction angle for 21 runs in both the compressional and tensional regime. The constant and uniform elemental resolution of the runs varies from $256 \times 64$ to $1,024 \times 256$ elements. Lower resolution runs were performed, but do not resolve the viscous inclusion well ($\leqslant 2$ elements) and are not shown (see instead Fig. 12 of Kaus, 2010). We monitor the relative residual as a measure of convergence, but the number of Nonlinear Iterations (NI, see Section 2.1.2) is fixed at 1,000. The red symbols in Fig. 5 indicate runs for which the residual is not monotonously decreasing (after the first peak in residual), as is evident from Fig. 6 (black versus red lines). In fact, the higher the internal friction angle, the more iterations are needed to reach a particular residual tolerance, and Fig. 6 also shows that higher internal friction angle runs stall at higher residuals. This coincides with a greater deviation from the theoretical Coulomb solution. Note that in these higher angle runs multiple shear bands are generated and an asymmetry between the right and left shears develops, as shown in Fig. 7. The spurious shear bands can occur mainly at the top of the domain, as several curved pieces forming one shear band or as complete additional shears. Despite these difficulties, there is a clear trend of measured angles verging from Arthur to Coulomb angles for an increasing resolution. For example, the average deviation from the





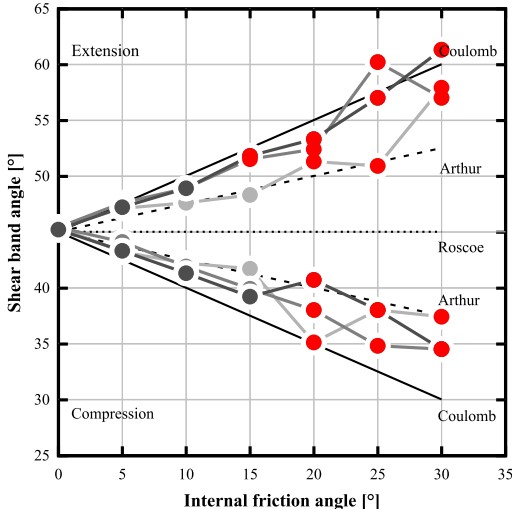

**Figure 5.** Measured shear band angle versus angle of internal friction for models in tension and compression. Resolution runs from $256 \times 64$ (light grey line), to $512 \times 128$ (grey) to $1,024 \times 256$ elements (dark grey). All models were run for $1,000$ NI; red symbols indicate runs with convergence behavior that is not monotonous. The black lines represent the theoretical angles of Coulomb (solid), Arthur (dashed) and Roscoe (dotted). We have corrected one of the automated shear band angle measurements manually – that of the $1024 \times 256$ el. extension case with $\phi = 25°$ – because it was computed using two different shear bands.

theoretical Coulomb angle decreases from $2.8°$ to $0.8°$ in extension when going from a resolution of $256 \times 64$ elements to $1,024 \times 256$ elements.

### 3.2.3 Discussion

As brittle failure in rocks is more appropriately described by pressure-dependent plasticity than by the perfectly-plastic defor-
mation (Gerbault et al., 1998) used in the punch problem, our material model plugin includes frictional plasticity. Testing this
pressure-dependent plasticity with the brick benchmark, shear band angles were found to increase with internal friction angle
$\phi$ as expected, almost all falling within the theoretical values of Arthur and Coulomb. Moreover, with increasing mesh resolu-
tion, the angles approach the Coulomb theoretical angle $\theta = 45 \pm \frac{\phi}{2}$ and the variation in error with respect to Coulomb angles
decreases. This tendency towards Coulomb angles for higher mesh resolution was also reported by Lemiale et al. (2008), Kaus
(2010) and Buiter (2012), because at higher resolution the viscous inclusion is better resolved. Kaus finds that at least 10 to 20
elements are required horizontally within the inclusion to obtain Coulomb angles. This corresponds to our two highest resolu-
tions. Choi and Petersen (2015) recently showed that consistent Coulomb angles can be achieved by an (initially) associated
flow law (where $\phi = \psi$).

Interestingly, internal angles of friction larger than $15°$–$20°$ lead to irregular convergence behavior that stalls at higher
relative residuals; these runs also show shear band angles further away from the theoretical Coulomb angle and multiple





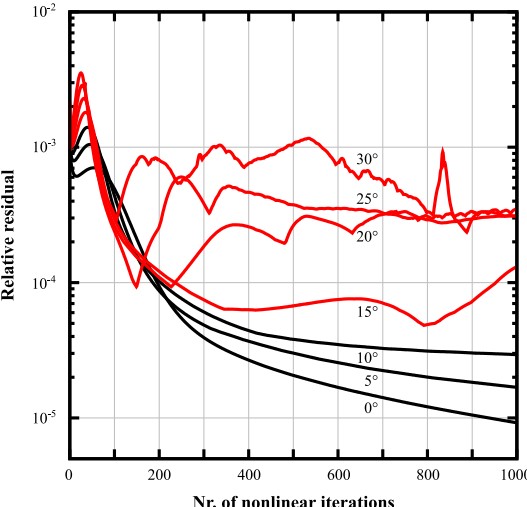

**Figure 6.** Measured relative residual versus the number of nonlinear iterations for models of extension. Elemental resolution is $512 \times 128$ elements. Black lines represent runs with well-behaved convergence.

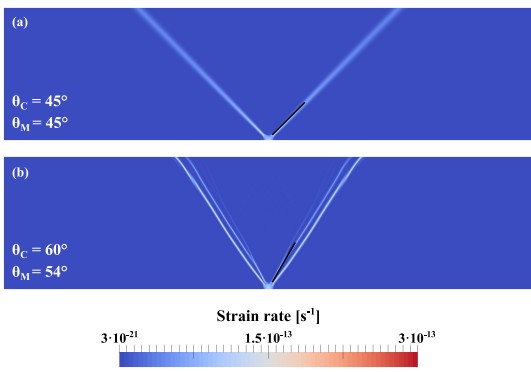

**Figure 7.** Strain rate fields for (a) $\phi = 0°$ and (b) $\phi = 30°$ for a $512 \times 128$ elemental resolution. The models were run in extension for $1,000$ NI. Black lines indicate the theoretical Coulomb angle $\theta_C$. Measured shear band angles $\theta_M$ are also given.

additional shears. These additional shears often do coincide with the theoretical angle, see for instance Fig. 7b. Why these latter shears are not dominant (highest strain rate) would require further investigation. Note that the models of Kaus (2010) also show multiple shear bands and that these spurious bands and stalling of convergence are shown to be the result of the dynamic pressure dependence of the Drucker–Prager yield criterion by Spiegelman et al. (2016).

5    However, we have shown that we consistently obtain shear band angles between Arthur and Coulomb theoretical angles at sufficient resolution and that these angles converge to Coulomb angles with increasing resolution. This despite the fact that



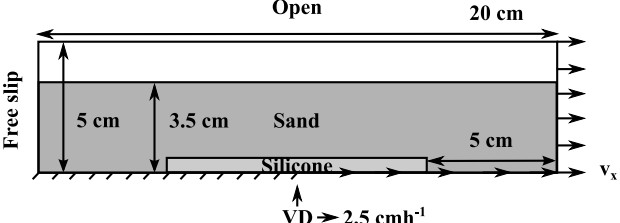

**Figure 8.** The sandbox experiment set-up after Buiter et al. (2006). VD: Velocity Discontinuity moving at the same speed as is prescribed on the right and bottom boundary. Left of the VD, the bottom boundary is no slip. The left vertical boundary is set to free slip, while the top is open. The silicone layer measures $10 \times 0.5$cm.

our implementation is relatively basic: it does not include softening of cohesion or of the internal angle of friction as in Kaus (2010) and Buiter (2012), nor does it have a sophisticated guess of the initial stress state (Lemiale et al., 2008) or an incremental build-up of the prescribed boundary velocity (Kaus, 2010).

### 3.3 The sandbox extension experiment

Buiter et al. (2006) compared numerical and analog models of shortening and extension. We reproduce the numerical sandbox extension experiment, which was originally run with 6 different numerical codes and compared to the analog results described in Schreurs et al. (2006). This experiment has previously been repeated by Thieulot (2011) and – in a symmetrical version – by Gerya et al. (2013).

### 3.3.1 Model set-up

The analog sandbox Buiter et al. (2006) consists of a basal layer of weak, viscous silicone overlain by brittle sand. The sand is extended by the movement of the right vertical wall and the connected basal plate extending from the wall to, initially, the centre of the domain. We model this set-up with 3 compositional fields (Fig. 8): 1) a viscous basal layer, 2) an overlying Drucker–Prager dynamic pressure-dependent plastic sand layer and 3) a low-viscous sticky-air layer (Crameri et al., 2012) on top. Extension is driven by a prescribed horizontal velocity on the right vertical boundary and the right half of the lower

boundary, mimicking the effect of the moving basal sheet. Basal friction is not taken into account. This approach is appropriate, since Buiter et al. (2006) have shown that the nature of the basal contact is less important than the interaction of the velocity discontinuity (VD in Fig. 8) and the silicone. The rest of the bottom boundary has zero velocity (without smoothing of the velocity discontinuity) and this no slip area increases as the velocity discontinuity moves to the right of the domain. The left boundary is free slip and the top boundary open. Adaptive Mesh Refinement (AMR) is applied based on the effective strain

rate field to obtain a maximum local refinement of $0.39 \times 0.39$mm along the shear bands (Fig. 10). The model is run until $2$cm of extension has occurred, equalling $2,880$s of model time. Material properties and other model parameters are listed in Table 5.



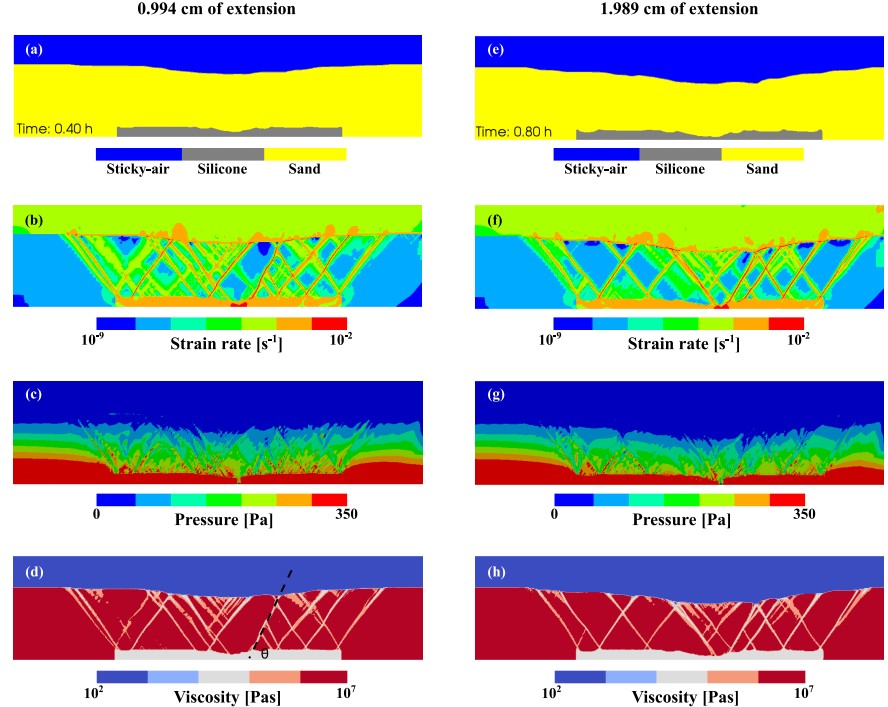

**Figure 9.** Results after $\sim 1$cm and $\sim 2$cm of extension of the numerical sandbox. (a) & (e): The three compositional fields. Note the asymmetric depression of the sand surface. (b) & (f): Strain rate field. (c) & (g): Total pressure field. (d) & (h): Viscosity field.

### 3.3.2 Model results

The results for $1$ and $2$cm of extension of the sandbox are presented in Fig. 9. The evolution of the model closely resembles what was found by Buiter et al. (2006). The initially symmetric system forms two conjugate shear zones stemming from the velocity discontinuity imposed by the velocity boundary conditions. With ongoing extension, the silicone layer distributes

5     deformation and the shear bands spread to the edges of the layer. After $1$ and $2$cm of extension, the angle of the shear bands left and right of the velocity discontinuity (see Fig. 9d) measure $\sim 51°, \sim 62°$ and $\sim 54°, \sim 60°$. With time the system becomes more and more asymmetric (compare left and right column of Fig. 9). The left side of the domain is at rest, while the outer right footwall moves at the prescribed velocity and we observe that the sticky-air properly accommodates the movement of the sand.

10     The viscosity field (Fig. 9h) is very irregular and displays sharp gradients up to 7 orders of magnitude. Figure 10 demonstrates the strain rate based AMR: refinement is localized in the low-viscosity shear bands, following the evolution of deformation. Through AMR, the total (velocity, pressure, temperature, composition) number of DOFs is limited to on average $\sim 475,000$ DOFs, instead of the $\sim 1,383,000$ DOFs ($\sim 527,000$ velocity DOFs) for a uniform resolution, thus decreasing the required computational resources by half (for the same number of cores).




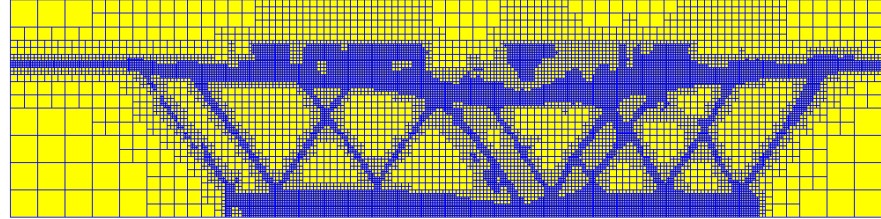

**Figure 10.** Numerical grid after $\sim 2$cm of extension of the numerical sandbox. Adaptive mesh refinement and coarsening based on the strain rate and density leads to an elemental resolution varying from $512 \times 128$ to $32 \times 8$ elements.

### 3.3.3 Discussion

The evolution of the numerical sandbox model – a model with AMR, high viscosity contrasts, large deformation and complex boundary conditions – compares well with those shown in Buiter et al. (2006) and Thieulot (2011). Although the right shear band angles of $62°$ and $60°$ after 1 and 2cm of deformation fall just outside the ranges found by Buiter et al. (2006) and

Thieulot (2011) of $45–55°$ and $45–53°$ respectively, they lie within the theoretical Arthur–Coulomb angles of $54–63°$ for a friction angle of $36°$ (Vermeer (1990), see also Section 3.2). Even when considering that the codes in Buiter et al. (2006) add strain softening by decreasing the friction angle from $36°$ to $31°$, for which the range of Arthur–Coulomb angles would be $52.75–60.50°$, their measured angles lie mostly below the Arthur angle. In the previous section, we demonstrated that our shear band angles fall within the Arthur–Coulomb range, and converge to Coulomb angles with increasing mesh resolution.

Differences in the measured angle of shear bands are therefore not surprising. A lower resolution run (maximum of $256 \times 64$ elemental resolution, not shown) results in angles to the right of the discontinuity of $60°$ and $55°$.

Similar to Buiter et al. (2006), Thieulot (2011) and Buiter (2012), we observe an increase in the number of shear bands with resolution as well as a decrease in their width. These are numerical effects tied to finite element models that should be taken into consideration when interpreting and comparing model results.

### 3.4  The slab detachment benchmark

Slab detachment, or break-off, in the final stage of subduction is often invoked to explain geophysical and geological observations such as tomographic images of slab remnants and exhumed ultra high pressure rocks (see for example Wortel and Spakman (2000) and references therein). Due to the increased interest in the process of slab tearing, it has recently been the subject of several numerical modeling studies (e.g. Gerya et al., 2004; Andrews and Billen, 2009; Burkett and Billen, 2009,

2010; van Hunen and Allen, 2011; Duretz et al., 2012, 2014). Numerical modeling of slab detachment is computationally challenging due to the mesh-resolution dependency of the strain (van Hunen and Allen, 2011) and the gradual decrease in monitoring particle density, the gradual overlapping of level sets (Hillebrand et al., 2014), or the gradual thinning of compositional fields (as is the case here) in the detachment area. Here we test ASPECT with the slab detachment model of Schmalholz




(2011), which considers a simplified geometry of detachment by viscous necking of a vertical lithospheric slab of nonlinear rheology in a linearly or nonlinearly viscous mantle. It has been extended to 3D by von Tscharner et al. (2014).

### 3.4.1 Model set-up

The 2D detachment model geometry is outlined in Fig. 11; both the lithosphere and the mantle are represented by a composi-
tional field. Schmalholz (2011) prescribes a nonlinear viscosity in the subducting lithosphere given by:

$$\eta_L = \eta_0 \dot{\epsilon}_e^{\left(\frac{1}{n}-1\right)} \tag{19}$$

where $\eta_0 = 4.75 \cdot 10^{11} \mathrm{Pas}^{\frac{1}{n}}$ and $n = 4$. Conversion to Eq. (5) results in the parameters listed in Table 6. Here we only reproduce the constant mantle-viscosity case of Schmalholz (2011), with $\mu_{mantle} = 10^{21} \mathrm{Pas}$. We use a set of 20 tracers placed on the outline of the slab to track the width and depth of necking. The necking width and depth, as well as time, are normalized by
the initial slab width of $80 \mathrm{km}$ and the characteristic time $t_c = 7.1158 \cdot 10^{14} \mathrm{s}$, respectively (Schmalholz, 2011).

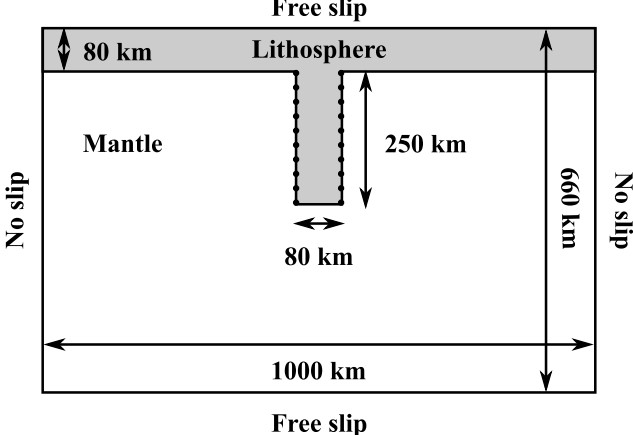

**Figure 11.** The detachment benchmark model set-up of Schmalholz (2011): a symmetric system of nonlinear viscous lithosphere (Eq. (19)) with a vertical slab extending into a linear viscous mantle. The top and bottom boundary are free slip, while the vertical boundaries are no slip. 20 Passive tracers are placed along the outline of the slab.

### 3.4.2 Results and comparison

The evolution of the detachment model is shown in Fig. 12: After about $20 \mathrm{My}$, the slab is fully necked and detached. Although a thin line of lithosphere composition is still visible, its value is less than $0.5$ and, thus, the infinite norm ignores this contribution to the viscosity, allowing for full detachment. Upon detachment, slab pull is removed and, thus, viscosity reaches $\mu_{max}$
throughout the remaining lithosphere (Fig. 12e&f).

It can be seen from the graph in Fig. 13 that the width of the necked zone through time agrees very well with the results of Schmalholz (2011). Only after $t = 0.6$, our results deviate slightly, showing faster necking. This could be the result of the




different averaging technique used here, as the infinite norm ignores the higher-viscosity contribution of the slab with respect to the other averaging methods (see Eq. (15)–(18) and Appendix A).

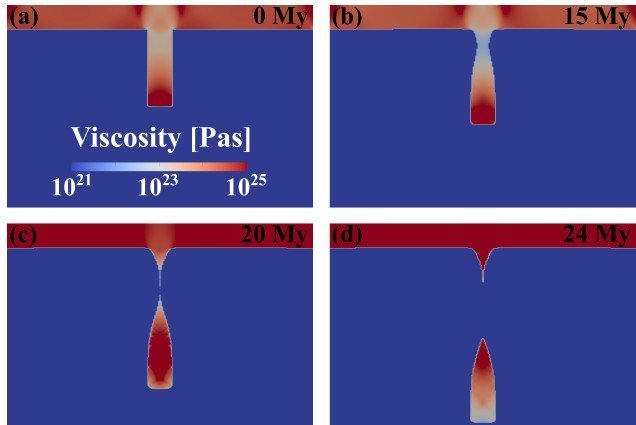

**Figure 12.** Detachment benchmark model evolution showing the viscosity field over time. After about 20My, necking is complete and the remaining lithosphere reaches a high, uniform viscosity.

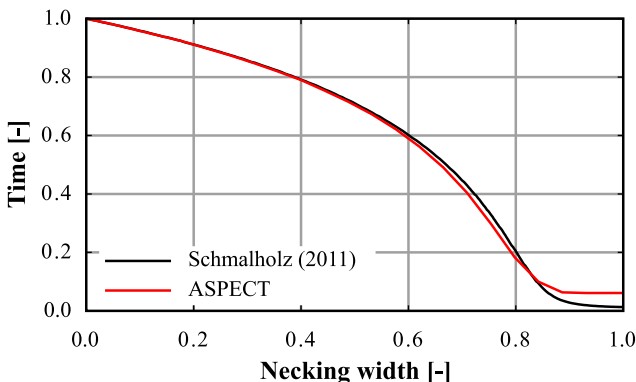

**Figure 13.** Nondimensional necking width versus time for ASPECT and Schmalholz (2011). The ASPECT necking width is calculated from the 20 tracer positions. Because the tracers above the necking zone no longer move after detachment, the thus calculated width stagnates after t $\simeq$ 0.9.

### 3.4.3 Discussion

The three previous benchmarks focussed on plastic rheologies. The detachment benchmark involves modeling of a highly
5   nonlinear, power-law viscosity, which is often used in subduction modeling. Our observed model evolution compares well with that of Schmalholz (2011) and other codes (Hillebrand et al., 2014; Thieulot et al., in prep.). It also demonstrates the

effective splitting of a compositional field into two bodies and how the rheology interacts with the compositional fields through localisation of deformation at the slab hinge. It should be noted that for this type of rheology the solution is mesh-resolution dependent as well, and iterative convergence should be monitored as for plastic rheologies. Differences in model evolution can also arise from the particular viscosity averaging method applied, as will be discussed in Thieulot et al. (in prep.), who present

a 3D detachment benchmark based on Schmalholz (2011).

## 4    Viscoplastic subduction models

Lastly, we consider a geodynamical application of the implemented viscoplastic rheology: the spatiotemporal evolution of 3D subduction. So far, no ASPECT applications to 2D or 3D regional subduction have been published. To demonstrate ASPECT's promise in this field, we here present 3D models of free-plate intraoceanic subduction. Recent 3D subduction models have been

applied in the study of along-strike effects such as oblique convergence (Malatesta et al., 2013), toroidal flow (Schellart and Moresi, 2013), varying lithospheric structure (Mason et al., 2010; van Hunen and Allen, 2011; Capitanio and Faccenda, 2012; Duretz et al., 2014), slab width (Stegman et al., 2006; Schellart et al., 2007; Stegman et al., 2010) and the presence of lateral plates (Yamato et al., 2009). Four-dimensional (3D plus time) modeling allows us to investigate more realistically the generics of subduction (e.g. Crameri and Tackley, 2014; Chertova et al., 2014) as well as very specific regional problems (e.g. Capitanio

and Replumaz, 2013; Chertova et al., 2014; Sternai et al., 2014).

### 4.1    Model set-up

We discuss two models; the first is an adaptation of the free-plate model of Schellart and Moresi (2013) and considers no temperature effects and features constant viscosities except for a viscoplastic crustal layer. The second model is an extension of the first, where we add a temperature field and a temperature, pressure and strain rate dependent viscosity, resulting in a

nonlinear viscoplastic thermomechanically coupled system.

#### 4.1.1    Model 1

The first model comprises two free plates: an Overriding Plate (OP) and a Subducting Plate (SP) (see Fig. 14). The SP is made up of a crustal layer of non-frictional (von Mises) viscoplastic rheology and a mantle lithospheric layer of constant viscosity (see Table 8 for actual values). The tip of the SP extends into the mantle for 200km. The OP is of one composition and has a

constant viscosity, as does the surrounding mantle. These four compositions are each represented by a compositional field.

#### 4.1.2    Model 2

The second model augments the first with an Adjacent Plate (AP) separated from the other plates by a 20 km thick Weak Zone (WZ). The SP and OP are extended and their thickness is no longer fixed, but based on the temperature field, as if they originate from ridges situated at the left and right vertical domain boundaries. The initial temperature distribution (Fig. 15) in the plates

is computed according to the age-based plate cooling model, for a mantle temperature $T_a$ of $1,593$K and surface temperature



$T_0 = 293$K (Eq. (4.2.24) of Schubert et al. (2001), $\kappa = 10^{-6} \, \text{m}^2\text{s}^{-1}$). Thickness of the plate is defined at the temperature $T$ for which $\frac{T_a - T}{T_a - T_0} = 0.1$. The maximum thickness of the plate (for time to infinity) is set to 125km, based on Schubert et al. (2001). At the trench, both plates have the same thickness as in model 1. The Adjacent Plate (AP) has a fixed thickness of 100km for an age of $\sim 60$My. From a depth of 125km, a linear temperature increase is prescribed everywhere to a bottom temperature

5  of 1,771K. The WZ and AP are of constant viscosity, but all other compositions are of nonlinear rheology, with which model 2 differs strongly from model 1. The specific rheological parameters for diffusion and dislocation creep and Drucker–Prager plasticity can be found in Table 8.

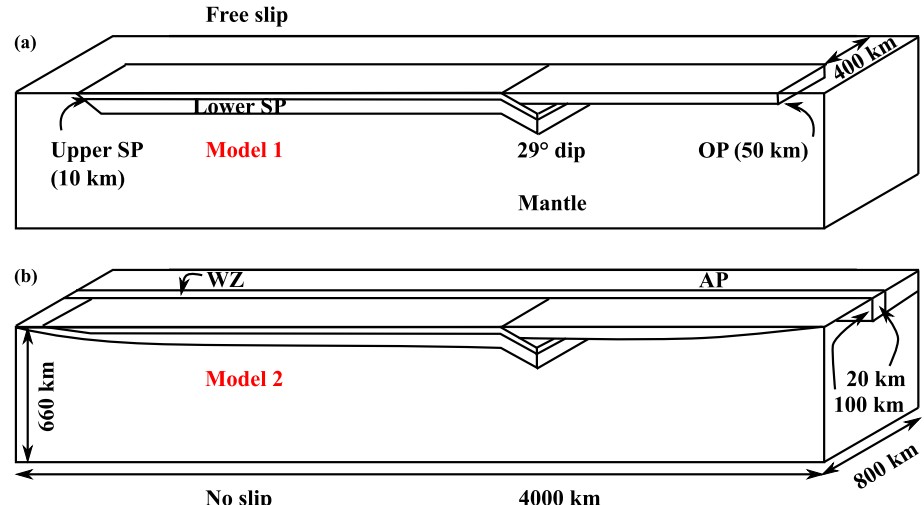

**Figure 14.** 3D Subduction model set-ups (also, see Table 7). (a) Model 1: a free Overriding Plate (OP) and a Subducting Plate (SP) with a trench at $x = 2,400$km. At start-up, the slab extends 200km into the mantle (measured vertically) at an angle of $29^\circ$. The 100km thick SP consists of 2 compositional layers, the OP of only 1 composition. All boundaries are free slip, except the no slip bottom boundary. (b) Model 2: A 58.8My old Adjacent Plate (AP) is added, separated from the OP and SP by a Weak Zone (WZ) of 20km width. The initial temperature distribution of the SP and OP is based on the plate cooling model dependent on age, which increases from the ridge situated at the left, respectively right, vertical domain boundary, up to an age of 16My for the OP and 60My for the SP, resulting in thicknesses of 50km and 100km at the trench, respectively. An adiabat of $0.25$Kkm$^{-1}$ is prescribed in the mantle. The bottom temperature is fixed at 1,728K, the top at 293K.

## 4.2 Results

### 4.2.1 Model 1

10  Figure 17 depicts the evolution of the subduction system of model 1 over time. The pull of the slab extending into the mantle results in a plastically-weakened subduction fault zone through high strain rates. Although this allows for decoupling from the surface, mechanical coupling is strong enough for the OP to move towards the trench. There the OP thickens and a small





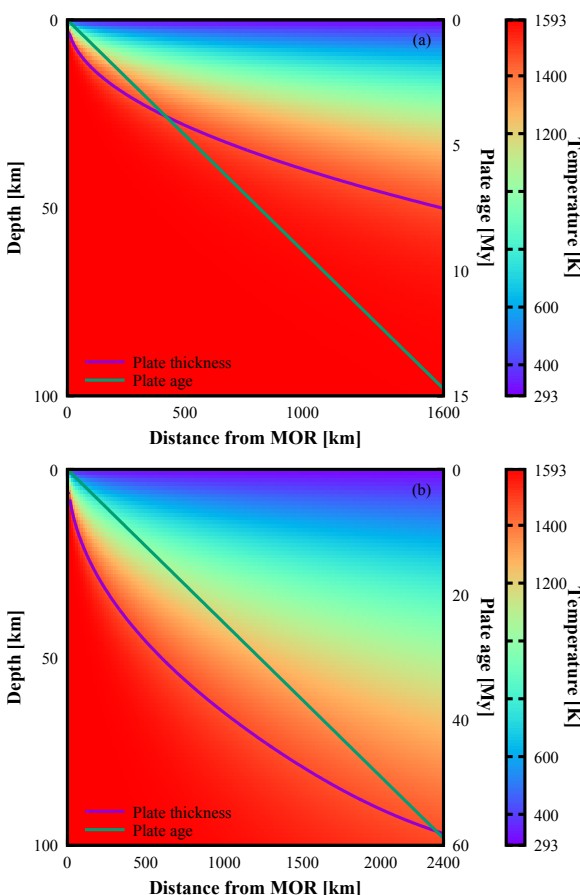

**Figure 15.** Temperature distribution with depth and distance from the ridge for (a) the OP and (b) the SP.

portion of OP material is entrained with the slab. Within the first ∼ 7My, the velocity of the plates steadily increases about 1 order of magnitude and reaches several centimeters per year. Similar to Schellart and Moresi (2013), poloidal and toroidal flow can be observed close to the slab; flow into the trench alters the shape of the plates. After 8.5My, plate velocities drop when the slab tip reaches a depth of 660km (bottom of the domain) and the steep slab starts to bend in to accommodate lateral

5   sliding over the 660km discontinuity. With ongoing subduction, the length of slab being pushed along the 660km discontinuity increases and plate velocities increase to pre-sliding levels. At 35My, the plate has completely subducted, lying flat at the 660km discontinuity, while the trench has retreated a total of ∼ 1,000km.

    The AMR based on composition and viscosity follows the outline of the plates as they move through the mantle (Fig. 17), resulting in a local resolution of roughly 3km while the mantle is resolved with ∼ 50km elements.



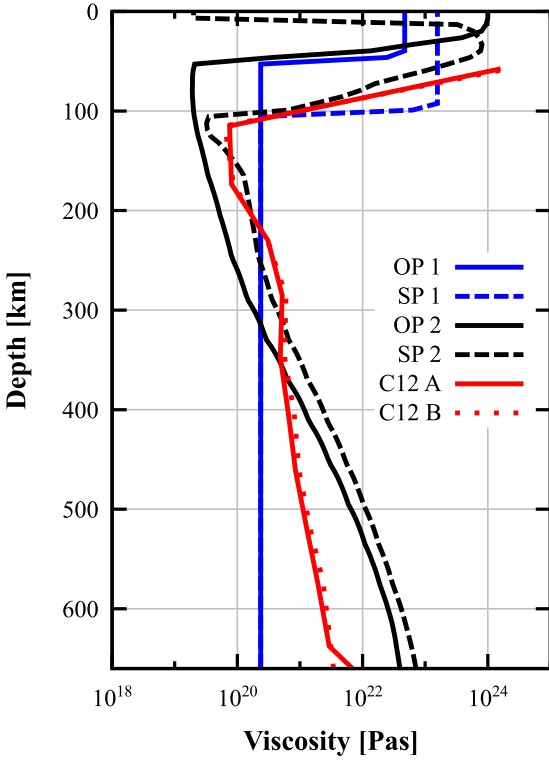

**Figure 16.** Viscosity profiles at $0$My for the OP (at $x = 2,700$km) and SP (at $x = 2,200$km) of model 1 and 2. For comparison, the viscosity profiles derived by Cizkova et al. (2012) are included.

#### 4.2.2 Model 2

The SP of model 2 steepens for the first $12$My (Fig. 18) until full-fledged subduction starts. Subduction is much slower than in model 1: while in model 1 subduction is completed by $35$My, rollback of the slab in model 2 only sets in around $50$My. Simultaneously with slab rollback, the subduction channel is weakened by asthenospheric inflow into the gap between the OP and SP. Formation of this gap is probably initiated by the fact that the OP does not completely release itself from the lateral boundary (see snapshot at $49$My). Even though the slab reaches the bottom boundary around $50$My and starts to shallow, it does not lie flat on the $660$km boundary, but continues to hover above it. Also note the halo of increased strain rate around the tip of the slab, and the small-scale strain rate features in the mantle compared to model 1.

#### 4.3 Discussion

The evolution of model 1 strongly resembles that of Schellart and Moresi (2013), on whose set-up the model is based: the slab first sinks freely until it reaches the $660$km bottom boundary and then starts draping while rolling back. The absence of folding


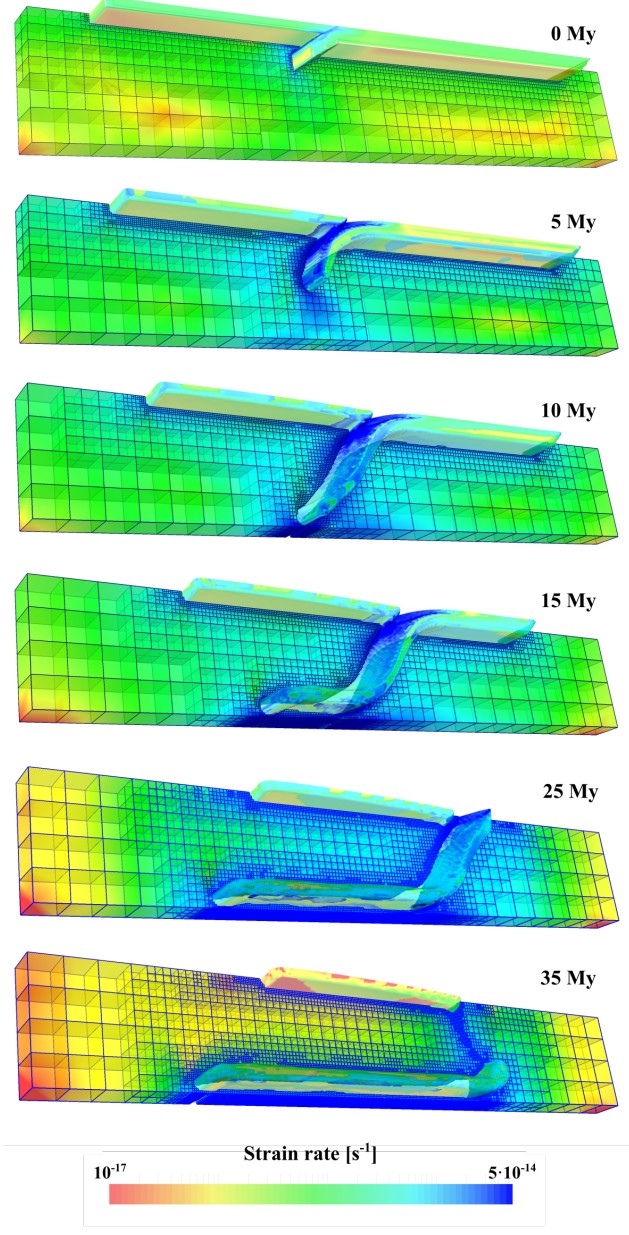

**Figure 17.** Model 1 – strain rate field over time together with SP and OP isocontours. Also shown is the adaptive mesh following the SP into the mantle.

of the slab at the $660\text{km}$ boundary and the differences in timing of the aforementioned events are probably due to the weak, linearly viscous layer of the SP that is left out here (and the lesser extent of the domain in the y-direction). This leaves the plate





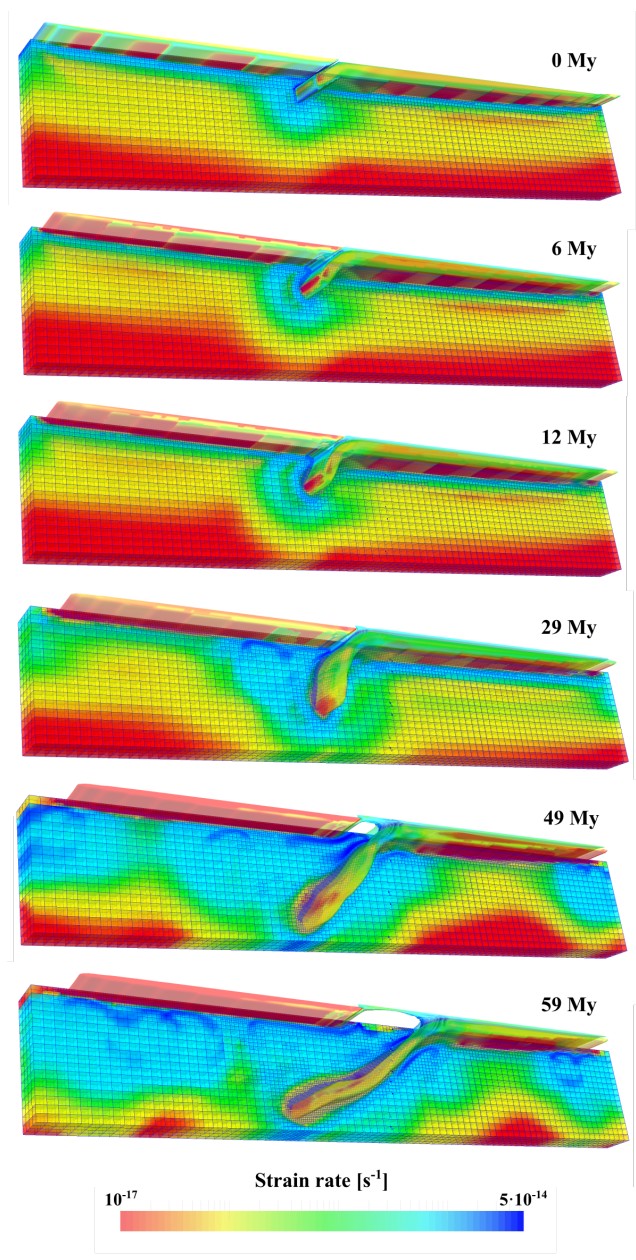

**Figure 18.** Model 2 – strain rate field over time together with SP, SP crust and OP isocontours. Also shown is the adaptive mesh following the SP into the mantle.

stronger and more resistant to folding than for Schellart and Moresi (2013). Note that our viscoplastic SP of model 2 also does not show draping at the 660km boundary.



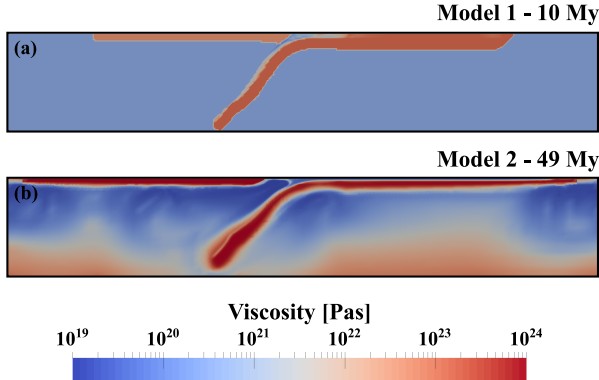

**Figure 19.** Viscosity snapshots of (a) model 1 and (b) model 2 at comparable moments in the respective subduction evolutions.

In switching from model 1 to this thermomechanically coupled model 2, we found changes to the set-up were necessary to avoid subduction of the plate at locations other than the slab tip (i.e. the sides and back of the plate would subduct as well) due to high mantle temperatures. Therefore, we added the adjacent plate and transform fault. By locating the plate ridges at the left and right vertical boundaries, free motion of the plates perpendicular to the trench is still enabled. Mesh resolution here was

reduced over time because the refinement strategy chosen focused on the compositional fields, which moved away from the boundaries. This increased the coupling of the OP plate to the left boundary unfortunately, limiting the plate's ability to move.

The subduction evolution of model 1 and 2 in Figs. 17 and 18 clearly differs. Models in the Appendix of Schellart and Moresi (2013) have shown that the addition of adjacent plates in itself does not affect the geometry of the SP over time (although velocities are affected). Therefore, the differences derive from the temperature, pressure, strain rate and composition

dependent rheology. Indeed, a snapshot (Fig. 19) of the viscosity field of the SPs and OPs shows that the viscosity of the plates of model 2 is about an order of magnitude higher (cut-off at $10^{24}$Pas). As both the SP and the OP of model 2 keep growing at the trench, they also experience more mantle drag than in model 1. Moreover, the model 2 slab tip is surrounded by a higher-viscosity area due to local cooling of the mantle. These rheological differences slowing down the subduction process are also evident from the strain rate in Figs. 17 and 18, showing a much weaker slab and mantle in model 1. A full investigation into

the differences between mechanical and thermomechanical viscoplastic models is beyond the scope of this paper.

More elaborate models of subduction should incorporate phase changes and latent heat effects as well as adiabatic and shear heating. This is also possible with ASPECT and we include an example of such a 2D model in Appendix B.

## 5    Discussion

The four benchmarks shown using our viscoplasticity implementations in ASPECT either reproduce the available analytical

solution (Section 3.1), or compare well with theory (Section 3.2) or the results of other codes (Sections 3.2–3.4, see also Tosi et al. (2015) for a benchmark with ASPECT using a different viscoplastic formulation). Thus validating our implementations,



they allowed us to set up a 4D model of oceanic subduction, exemplifying the functionality that our implementations have added.

It should be noted that although the rheology described in this paper is often applied in numerical modeling, more elaborate laws have been proposed. For example, Choi and Petersen (2015) argue that numerical models should incorporate an initially associated plastic flow rule that evolves into a non-associated flow rule with increased slip to assure persistent Coulomb shear band angles while avoiding unlimited dilatation. Another addition would be to include plastic softening or hardening – changes in the yield surface due to the accumulated strain. As of December 2016, the ASPECT developer version provides a way to use either tracer particles or compositional fields to track the strain and weaken both the cohesion and friction angle (ASPECT, 2016). Considering creep flow laws, improvements could be made by adding Peierls creep, a dislocation mechanism acting at low temperatures and/or high stresses (e.g. in parts of the slab Karato, 2008; Duretz et al., 2011; Garel et al., 2014). Other authors such as Farrington et al. (2014) and Fourel et al. (2014) have investigated the effect of incorporating elasticity into models of lithosphere subduction, demonstrating that although observables such as dip angle, slab morphology and plate motion are not affected, an elastoviscous or elastoviscoplastic rheology leads to different viscosities in the hinge of the slab.

Incorporating more realistic nonlinear rheologies such as described in this paper creates the necessity for additional nonlinear iterations within a single time step. Also, we have seen that at higher mesh resolutions, more of such iterations are required to converge the solution. This greatly increases model run time and therefore it is important to implement a more efficient nonlinear solving strategy than the Picard iterations currently used by ASPECT. The more efficient Newton iterations (see for example Popov and Sobolev, 2008; May et al., 2015; Rudi et al., 2015) could help prevent nonlinear rheologies forming a bottleneck. Such iterations have been implemented in ASPECT (Fraters et al., in prep.) and are currently being incorporated in the release version.

## 6   Conclusion and outlook

Numerical modeling of intricate geodynamic processes such as crust and lithosphere deformation and plate subduction encompasses challenges at different levels. For one, the 4D nature of the subduction process requires state-of-the-art numerical methods to efficiently handle the parallel computations necessary for such large problem sets. Secondly, models should incorporate realistic (non)linear rheologies to mimic nature as close as possible. Thus arises the need for algorithms that can solve highly nonlinear equations and deal with large viscosity contrasts effectively. Thirdly, far-field effects of mantle flow and plate motion cannot be ignored, and neither can topography building, resulting in a demand for complex boundary conditions such as open boundaries (Chertova et al., 2012) and free surfaces (Kaus et al., 2010; Crameri et al., 2012; Rose et al., 2017).

In this paper, we have shown that the open source code ASPECT is up to these challenges. Building on its modern, massively parallel numerical methods, we have outlined here our basic additions that enable viscoplastic crust and lithosphere modeling. We then tested and validated the algorithms with four different benchmarks well-known in the geodynamic modeling community. Last, we highlighted the possibilities arising from the adaptations with 4D thermomechanically coupled viscoplas-



tic models of interoceanic subduction showing that ASPECT is a serious contender in the field of lithospheric subduction modeling.

The continued development of ASPECT based on the needs of its expanding user and developer community ensures ev-ergrowing capacities and possibilities. Important recent additions are a full free surface (Rose et al., 2017), the formation and migration of partial melt (Dannberg and Heister, 2016) and active particles (Gassmöller et al., 2016, submitted). Also, the extensive user manual (Bangerth et al., 2016) accompanying all developments is a great asset for new and current users. In consequence, opportunities for future research are reinforced and a firm foundation is provided for ASPECT in the geodynamics community.

## 7   Code availability

The plasticity formulation has become part of the ASPECT distribution. A pull request for a plugin that calculates a constant lithostatic pressure profile needed for open boundary traction conditions is available from GitHub and will be incorporated in ASPECT. Input parameter files to reproduce the benchmarks will be incorporated as well.

## 8   Data availability

There is no data to be made available.

## Appendix A:  Self-consistent subduction and compositional averaging

As discussed in Section 2.2.2, the choice of averaging method for models of multiple compositions can significantly influence the results (Deubelbeiss and Kaus, 2008; Schmeling et al., 2008). Based on a number of experiments, we have chosen the infinite norm as method of choice for this paper. One of the models on which this choice is based, is that of Schmeling et al. (2008), on which we will elaborate below.

## A1   Model set-up

The 2D linear viscous model is composed of three compositions: the mantle, subducting lithosphere and sticky-air to allow for surface topography build-up and detachment of the lithosphere from the top boundary (Fig. 20, Table 9). The subducted part of the lithosphere supplies the force to start subduction. The four different averaging methods in Eq. (15)–(18)) are tested at different resolutions.

## A2   Model results

The evolution of subduction is summarized in a plot of the slab tip depth over time in Fig. 21. It is clear that the effect of averaging method dominates over that of resolution, but that both are significant. As do Schmeling et al. (2008) (shaded areas

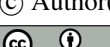



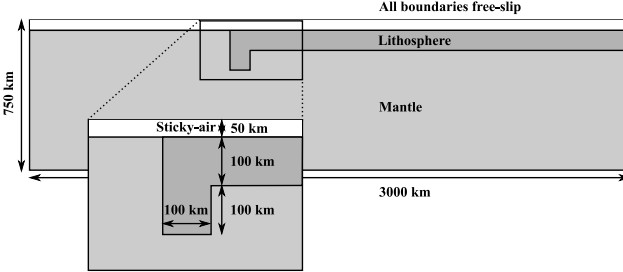

**Figure 20.** The self-consistent subduction benchmark model set-up. The mantle, subducting plate and sticky-air are represented by three compositional fields of constant viscosity.

in Fig. 21), we find that subduction is fastest for harmonic averaging and slowest for arithmetic averaging. For a given method, a higher resolution speeds up subduction, except for harmonic averaging. For an explanation of these results, see Schmeling et al. (2008). Comparison with the absolute results of Schmeling et al. (2008) is complicated by different computational methods, elements, minimum resolutions and mesh configurations, e.g. compare the dark solid and dotted red lines in Fig. 21.

Snapshots of the viscosity field are shown in Fig. 22: the infinite norm model's field shows the least artefacts from compositional under- and overshoot, the harmonically averaged model the most. Wall time for the first $2,000$ timesteps is reported for the highest resolution model of each averaging method in Table 10; the infinite norm is the most computationally expensive.

### A3    Choice of averaging method

The infinite norm selects the parameters of the field that is greatest in a specific point. It thus counteracts the numerical
diffusion of the compositional boundaries in the calculation of composition dependent parameters, but unfortunately also sharpens possible viscosity contrasts between the fields. This increases computational time. For more complex models where wall time is an important factor, we recommend using the geometric averaging method. ASPECT's more recent option to average viscosities over the quadrature points within each element can also decrease wall time significantly.

### Appendix B: Compressible subduction with phase changes, open boundaries and a free surface

With this example of 2D subduction, we highlight some of the more recent additions to ASPECT: compositional field reactions (which can be used to implement phase changes), the true free surface (Rose et al., 2017) and traction boundary conditions. These features are used to set up a model of thermomechanically coupled viscoplastic subduction in which the plate motions are either prescribed on the vertical boundaries, or, at a later stage, material is free to move in/out of the model domain. A





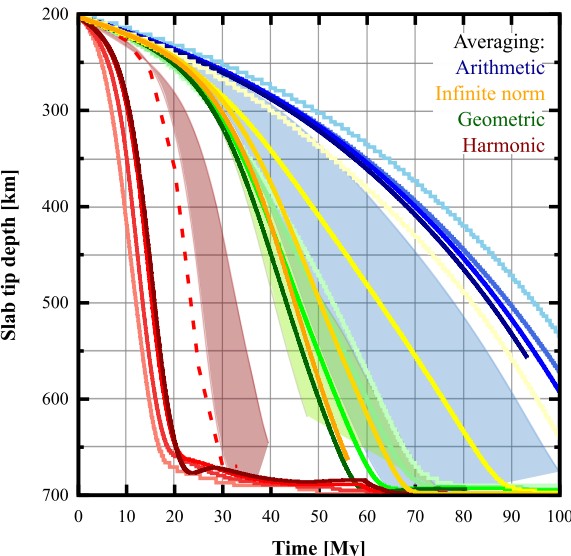

**Figure 21.** Slab tip depth versus model time for four different averaging methods of the contribution of the compositional fields to viscosity. Colors indicate the averaging method, while one color goes from light to dark with local resolution, which varies from $256 \times 64$ elements to $2,048 \times 512$ elements. Minimum resolution is always $128 \times 32$ elements. The dashed red line model has a resolution varying from $128 \times 128$ to $2,048 \times 2,048$ elements. Shaded areas represent results of Schmeling et al. (2008, Fig. 6).

compressible formulation of the governing equations is used, including shear heating, adiabatic heating and latent heat:

$$-\nabla \cdot (2\eta\dot{\epsilon}) + \nabla P = \rho \boldsymbol{g} \qquad (B1)$$

$$\nabla \cdot \boldsymbol{v} = -\beta\rho\boldsymbol{v} \cdot \boldsymbol{g} \qquad (B2)$$

$$\left(\rho c_p - \rho T \Delta S \frac{\partial X}{\partial T}\right)\left(\frac{\partial T}{\partial t} + \boldsymbol{v} \cdot \nabla T\right) \qquad (B3)$$

$$-\nabla \cdot k\nabla T = (2\eta\dot{\epsilon}) : \dot{\epsilon}$$

$$+ \alpha T \boldsymbol{v} \cdot \nabla P$$

$$+ \rho T \Delta S \frac{\partial X}{\partial P} \boldsymbol{v} \cdot \nabla P$$

where $\beta$ is the compressibility $\frac{1}{\rho}\frac{\partial \rho}{\partial P}$, $\rho = \rho_0(\beta(P - P_0))(1.0 - \alpha(T - T_0))$, $\Delta S = \gamma_t \frac{\delta_{\rho_t}}{\rho^2}$ and $\frac{\partial X}{\partial T} = -\gamma_t \frac{\partial X}{\partial P}$.

Phase changes (changes of one compositional field into another with different material properties) are implemented by extending our, now compressible, multicomponent viscoplastic material model with a depth-dependent transition function (e.g. Christensen and Yuen, 1985):

$$X = \frac{1}{2}\left[1.0 + \tanh\left(\frac{z - z_t - \gamma_t \frac{z_t}{P_t}(T - T_t)}{d\frac{z_t}{P_t}}\right)\right] \qquad (B4)$$





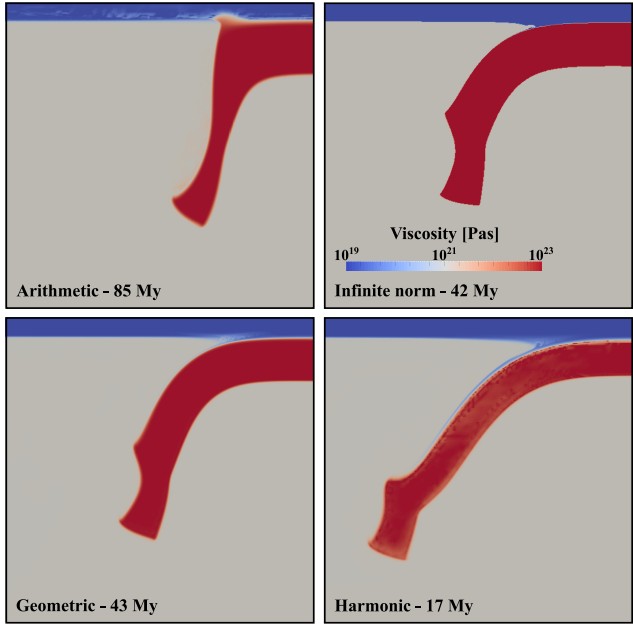

**Figure 22.** Viscosity field for each averaging method at $2,048 \times 512$ elements local resolution at similar moments in the subduction evolution.

$X$ represents the fraction of the new phase, $P_t$, $T_t$ and $z_t$ are the reference pressure, temperature and depth of the transition respectively and d the transition half-width in terms of pressure (assuming $P = P_{lith} = \rho g z$). The phase function derivatives in Eq. (B3) are computed as in the latent heat plugin of the *Material model* module.

Open boundary conditions are newly implemented as a plugin to the *Traction boundary conditions* module by prescribing the traction as Chertova et al. (2012):

$$\tau = -P_{lith}\hat{\boldsymbol{n}} \tag{B5}$$

with $\hat{n}$ the outward normal to the domain boundary. $P_{lith}$ is the lithostatic pressure calculated by numerical integration along the domain boundary of the density (i.e. the temperature and composition) of the previous timestep.

**B1  Model set-up**

The compressible subduction model considers ocean-continent subduction in a domain of $1,600$km by $1,000$km (Fig. 23). The model includes phase changes around $410$km and $660$km depth, see Table 11. An inward plate velocity is prescribed on the upper part of the left vertical boundary for the first $13$My at $4$cmyr$^{-1}$, while prescribed outward mantle velocities compensate for this volume increase. After $13$My, material is free to move in or out of the domain through open boundary conditions on the left boundary. On the right boundary, no slip conditions are switched to a prescribed velocity profile after $8$My with a plate inflow of $2$cmyr$^{-1}$. Apart from the subducting plate crust, which is of linear viscosity, all materials are nonlinear viscoplastic.





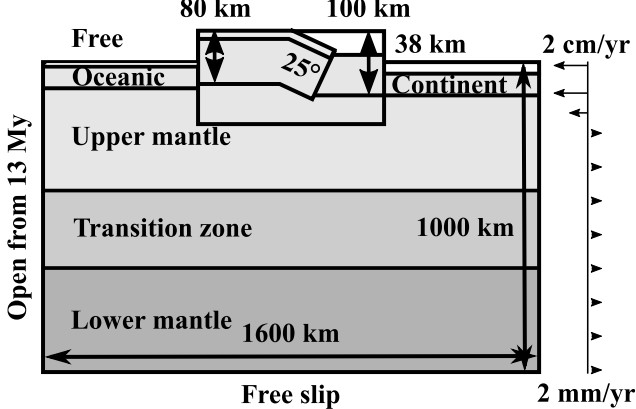

**Figure 23.** Compressible subduction model set-up: Subduction of an oceanic plate of 80km thickness underneath a 100km thick continental plate is initiated with an 80km slab. Different compositional fields are used to describe the oceanic and continental crust, and the upper mantle, transition zone and lower mantle material. A 2-times viscosity increase is also included at the 660km phase boundary. The top boundary is a true free surface, the right vertical boundary has a prescribed in/outflow as indicated. A similar flow ($4\mathrm{cmyr}^{-1}$ inflow) is prescribed on the left boundary until 13My, when the boundary is opened.

At 200km, crustal material is transformed to mantle material (as is done, for example, by Androvicova et al., 2013). Initial temperature is based on an adiabatic profile in the mantle and linear profiles in the plates.

## B2    Model results

Figure 24 shows the evolution of the compressible subduction model in terms of viscosity. When the left boundary is opened at 13My, just after the slab has reached the 410km phase boundary, sinking velocities increase to $10.5\mathrm{cmyr}^{-1}$. Upon reaching the 660km phase boundary (with a two-fold viscosity increase), the slab slows down to about $5\mathrm{cmyr}^{-1}$. The tip of the slab is impeded by the 660km boundary and moves at around $1\mathrm{cmyr}^{-1}$ along the boundary.

The change in flow through the left boundary is depicted in Fig. 25: When the left domain boundary is opened up by prescribing stresses instead of a fixed velocity profile, first of all a downward shift of the transition from in- to outflow is seen. Moreover, the subducting plate velocity initially increases up to $11\mathrm{cmyr}^{-1}$, together with an increase in transition and lower mantle velocity. When the slab reaches the 660km phase boundary around 16My, lower mantle outflow goes up rather uniformly throughout the lower mantle, but in/outflow above decreases. When the slab tip moves over the 660km boundary, all flow decreases in magnitude - gradually in the mantle and uniformly in the lower mantle.

The phase changes are clearly expressed in the density fields: the density isocontours in Fig. 24 show positive topography in the slab at the 410km discontinuity, while the 660km transition in the slab occurs deeper. The deformation of the surface at 18My shows a lowered subducting plate (about 1km below the initial top boundary), a topographic rise of continental crust above the subducting slab (of maximally 8km) and an elevated overriding continental plate (about 4km).





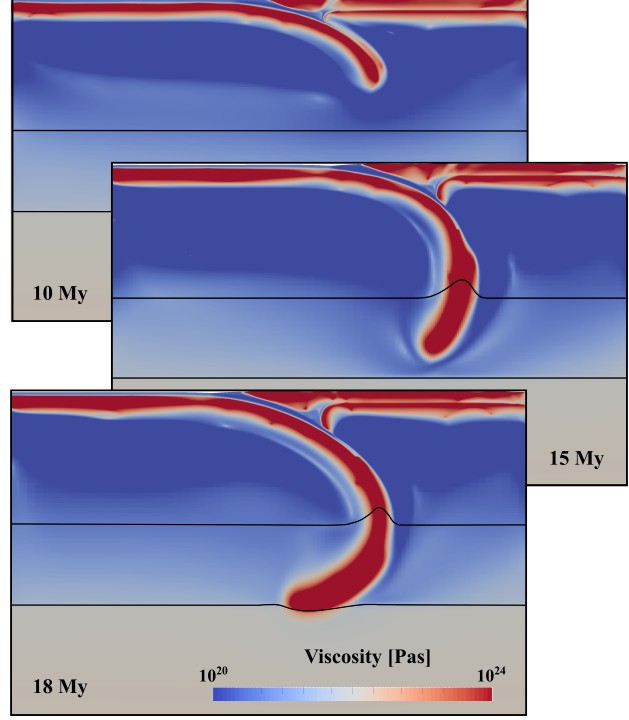

**Figure 24.** Compressible subduction evolution in terms of viscosity. Density is contoured at $3,700 \text{kgm}^{-3}$ and $4,200 \text{kgm}^{-3}$, demonstrating phase boundary topography in the slab.

## B3 Discussion and Conclusion

Figure 25 illustrates the forcing boundary conditions exert on subduction models: upon opening the left boundary a different flow pattern develops that changes over time in reaction to the internal dynamics of the system, i.e. it is more representative than the prescribed velocity profile. Together with the phase boundaries and free surface, such open boundary conditions allow

5    for more realistic models of subduction.

*Author contributions.* A. Glerum developed the code implementations and performed the simulations for this paper, with the exception of the 2D compressible subduction model in the Appendix. This latter model was constructed by C. Blom under supervision of C. Thieulot, A. Glerum and M. Fraters, based on code of M. Fraters (model set-up) and A. Glerum (open boundary conditions, rheology). C. Thieulot provided the model output of several benchmarks of his code ELEFANT for comparison and discussion. A. Glerum prepared the manuscript

10   with contributions from C. Thieulot and W. Spakman.

*Competing interests.* The authors declare that they have no conflict of interest.




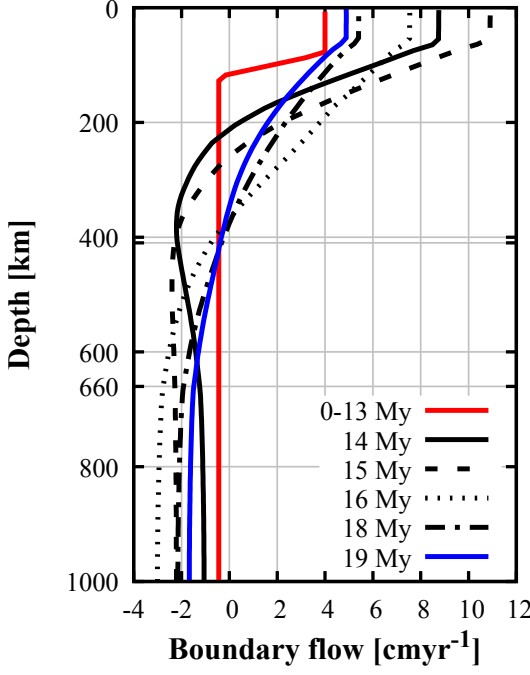

**Figure 25.** Left boundary in/outflow for the compressible subduction model. In red the velocity profile that is prescribed for the first 13My. In black the velocity profiles obtained with open boundary conditions. Positive values represent inflow, negative outflow.

*Disclaimer.* The authors declare that they have nothing to disclaim.

*Acknowledgements.* We thank W. Bangerth, T. Heister, R. Gassmöller and J. Dannberg for their precious help concerning the use and fine tuning of ASPECT. We also thank H. Schmeling for providing the data of his 2008 subduction benchmark paper and S. M. Schmalholz for the slab detachment data of his 2011 paper. A. P. van den Berg is thanked for constructive discussions. This work was funded by The Netherlands Research Centre for Integrated Solid Earth Science (grant number ISES-2012-89) and the Research Council of Norway through its Centres of Excellence funding scheme (project number 223272). ISES also provided financial support for the in-house computer on which computations were done. ASPECT is hosted by the Computational Infrastructure for Geodynamics (CIG), which is supported by the National Science Foundation award NSF-094946. We also thank CIG for support for A. Glerum, C. Thieulot and M. Fraters to attend the ASPECT Hackatons. We acknowledge the work of I. Rose (averaging of compositional fields), B. Myhill (dislocation and diffusion creep) and J. Naliboff (combining plastic yielding and creep) in implementing similar available algorithms into ASPECT during the writing of this article but after our running of the experiments. Figures were created using the open source software ParaView, Inkscape and gnuplot.

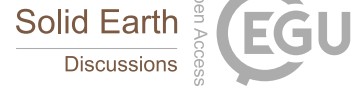

**Table 1.** Definition of symbols

| Parameter name | Symbol | Unit |
|---|---|---|
| Activation volume* | $V_{df|dl}$† | $\mathrm{m^3mol^{-1}}$ |
| Activation energy* | $Q_{df|dl}$ | $\mathrm{Jmol^{-1}}$ |
| Artificial diffusivity | $\nu_h$ | $\mathrm{m^2s^{-1}}$ |
| Burgers vector length | $b$ | $0.5 \cdot 10^{-9}$ m |
| Cohesion* | $C$ | Pa |
| Compositional field i | $c_i$ | - |
| Effective deviatoric strain rate | $\dot{\epsilon}_e$ | $\mathrm{s^{-1}}$ |
| Effective viscosity | $\mu_{eff}^{vsc|df|dl|cp|pl|vp}$ | Pas |
| Gas constant | $R$ | $8.314\mathrm{JK^{-1}mol^{-1}}$ |
| Grain size | $d$ | 0.01m |
| Grain size exponent | $m$ | - |
| Gravity vector | $\boldsymbol{g}$ | $\mathrm{ms^{-2}}$ |
| Initial effective strain rate | $\dot{\epsilon}_{init}$ | $\mathrm{s^{-1}}$ |
| Initial linear viscosity* | $\mu_{init}$ | Pas |
| Internal angle of friction* | $\phi$ | ° |
| NI convergence criterion† | $\epsilon_{\mathrm{u|p}}$ | - |
| Minimum/maximum viscosity | $\mu_{min|max}$ | Pas |
| Preexponential factor | $A_{df|dl}$ | $\mathrm{Pa^{-n}s^{-1}}$ |
| Prefactor* | $B_{df|dl}$ | $\mathrm{Pa^{-n}s^{-1}}$ |
| Reference density* | $\rho_0$ | $\mathrm{kgm^{-3}}$ |
| Reference temperature* | $T_0$ | K |
| Reference viscosity | $\mu_{ref}$ | Pas |
| Scaling factor* | $\beta_{df|dl}$ | - |
| Shear modulus | $K$ | 80 GPa |
| Specific heat* | $c_{\mathrm{p}}$ | $\mathrm{Jkg^{-1}K^{-1}}$ |
| Strain rate tensor | $\dot{\epsilon}$ | $\mathrm{s^{-1}}$ |
| Stress exponent* | $n$ | - |
| Temperature | $T$ | K |
| Thermal conductivity* | $k$ | $\mathrm{Wm^{-1}K^{-1}}$ |
| Thermal diffusivity | $\kappa = \frac{k}{\rho c_p}$ | $\mathrm{m^2s^{-1}}$ |
| Thermal expansivity* | $\alpha$ | $\mathrm{K^{-1}}$ |
| Time | $t$ | s |
| Total pressure | $P$ | Pa |
| Velocity vector | $\boldsymbol{u}$ | $\mathrm{ms^{-1}}$ |
| Viscosity | $\mu$ | Pas |
| Yield strength | $\sigma_y$ | Pa |

† Abbreviations: df. = diffusion, dl. = dislocation, vsc. = viscous, cp. = composite, pl. = plastic,

vp. = viscoplastic. NI = Nonlinear Iterations. **33**

* Material parameter specified per compositional field.



**Table 2.** Characteristics of performed experiments

| Benchmark | $nc^\dagger$ | Rheology | Time stepping | Solution | References |
|---|---|---|---|---|---|
| Indentor | 1 | Rigid plastic | no | Analytical | Kachanov (2004); Thieulot et al. (2008) |
| Brick | 2 | Linear viscous, frictional plastic | no | Theory + other codes | Lemiale et al. (2008); Kaus (2010) |
| Sandbox | 3 | Linear viscous, frictional plastic including sticky-air | yes | Other codes | Buiter et al. (2006); Thieulot (2011) |
| Detachment | 2 | Power-law viscous | yes | Analytical + other codes | Schmalholz (2011); Hillebrand et al. (2014) |

$\dagger nc$: number of compositions. None of the benchmarks include temperature effects in the rheology

**Table 3.** The indentor benchmark model parameters

| Parameter | Value [Unit] |
|---|---|
| Domain width | 1 |
| Domain height | 1 |
| Resolution | $512 \times 512$ el. |
| Gravitational acceleration | 0 |
| Reference viscosity $\mu_{\mathrm{ref}}$ | $10^3$ |
| Minimum viscosity $\mu_{\mathrm{min}}$ | $10^{-3}$ |
| Maximum viscosity $\mu_{\mathrm{max}}$ | $10^3$ |
| Capped viscosity $\mu_{eff}$ | Eq. (13) |
| Stokes solver tolerance | $10^{-9}$ |
| Nr. of nonlinear iterations NI | 500 |
| Surface pressure normalization | no |
| Indentor width $p$ | 0.125 |
| Indentor velocity $v_p$ | 1.05 |
| Nr. of cores | 24 |
| DOF/core | $142,294$ |
| Wall time | $\sim 6$h |
| **Rigid plastic medium** | |
| Density $\rho$ | 0.01 |
| Initial viscosity $\mu_{\mathrm{init}}$ | $10^1$ |
| Cohesion $C$ | 1 |
| Angle of internal friction $\phi$ | $0°$ |





**Table 4.** The brick benchmark model parameters

| Parameter | Value [Unit] |
|---|---|
| Domain width | 40km |
| Domain height | 10km |
| Resolution | $128 \times 32$–$1024 \times 256$ el. |
| Gravitational acceleration | $10\text{ms}^{-2}$ |
| Applied horizontal velocity $v_x$ | $2 \cdot 10^{-11}\text{ms}^{-1}$ |
| Reference viscosity $\mu_{\text{ref}}$ | $10^{24}\text{Pas}$ |
| Effective viscosity $\mu_{eff}^{vp}$ | Eq. (12) |
| Minimum viscosity $\mu_{\text{min}}$ | $10^{19}\text{Pas}$ |
| Maximum viscosity $\mu_{\text{max}}$ | $10^{26}\text{Pas}$ |
| Capped viscosity $\mu_{eff}$ | Eq. (13) |
| Stokes solver tolerance | $10^{-7}$ |
| Nr. of nonlinear iterations NI | $10^3$ |
| Nr. of cores | 28 |
| Wall time | 10min–148h |
| **Viscoplastic medium** | |
| Constant density $\rho$ | $2,700\text{kgm}^{-3}$ |
| Initial viscosity $\mu_{\text{init}}$ | $10^{23}\text{Pas}$ |
| Linear viscous viscosity $\mu$ | $10^{25}\text{Pas}$ |
| Cohesion $C$ | 40MPa |
| Angle of internal friction $\phi$ | 0–30° |
| **Viscous inclusion** | |
| Constant density $\rho$ | $2,700\text{kgm}^{-3}$ |
| Linear viscous viscosity $\mu$ | $10^{20}\text{Pas}$ |





**Table 5.** The sandbox experiment model parameters

| Parameter | Value [Unit] |
|---|---|
| Domain width | 20cm |
| Domain height | 5cm |
| CFL number | 0.5 |
| Local resolution | $512 \times 128$–$32 \times 8$ el. |
| Applied horizontal velocity $v_x$ | $2.5\mathrm{cmh}^{-1}$ |
| Gravity acceleration | $9.81\mathrm{ms}^{-2}$ |
| Reference viscosity $\mu_{\mathrm{ref}}$ | $10^7\mathrm{Pas}$ |
| Initial effective strain rate $\dot{\epsilon}_{init}$ | $10^{-6}\mathrm{s}^{-1}$ |
| Effective viscosity $\mu_{eff}^{vp}$ | Eq. (11) |
| Minimum viscosity $\mu_{\mathrm{min}}$ | $10^2\mathrm{Pas}$ |
| Maximum viscosity $\mu_{\mathrm{max}}$ | $10^9\mathrm{Pas}$ |
| Viscosity capping | Eq. (13) |
| Nr. of NI per time step ($t_0$;rest) | 100; 20 |
| Stokes solver tolerance | $10^{-7}$ |
| Model end time | $3,100$s |
| Nr. of cores | 24 |
| DOF/core | $\sim 32,000$ |
| Wall time | 27h |

**Sticky-air**

| | |
|---|---|
| Constant density $\rho$ | $10\mathrm{kgm}^{-3}$ |
| Linear viscous viscosity $\mu$ | $10^2\mathrm{Pas}$ |

**Sand**

| | |
|---|---|
| Constant density $\rho$ | $1,560\mathrm{kgm}^{-3}$ |
| Linear viscous viscosity $\mu$ | $10^{13}\mathrm{Pas}$ |
| Cohesion $C$ | 10Pa |
| Angle of internal friction $\phi$ | $36°$ |

**Silicon**

| | |
|---|---|
| Constant density $\rho$ | $965\mathrm{kgm}^{-3}$ |
| Linear viscous viscosity $\mu$ | $5 \cdot 10^4\mathrm{Pas}$ |

Parameters are on sandbox scale, as they are used in the model.





**Table 6.** The detachment benchmark model parameters

| Parameter | Value [Unit] |
| --- | --- |
| Domain width | $1,000$km |
| Domain height | $660$km |
| CFL number | $0.5$ |
| Local resolution | $64 \times 64$–$256 \times 256$ el. |
| Gravity acceleration | $9.81 \mathrm{ms}^{-2}$ |
| Reference viscosity $\mu_{\max}$ | $3 \cdot 10^{22}$Pas |
| Minimum viscosity $\mu_{\min}$ | $10^{21}$ Pas |
| Maximum viscosity $\mu_{\max}$ | $10^{25}$ Pas |
| Viscosity capping $mu_{eff}$ | Eq. (13) |
| Max. nr. of NI per timestep | $50$ |
| Stokes solver tolerance | $10^{-5}$ |
| Model end time | $25$My |
| Temperature polynomial degree | $1$ |
| Nr. of cores | $28$ |
| DOF/core | $\sim 16,300$ |
| Wall time | $16$h |
| **Lithosphere** | |
| Constant density $\rho$ | $3,300 \mathrm{kgm}^{-3}$ |
| Activation volume $V$ | $0 \mathrm{m}^3 \mathrm{mol}^{-1}$ |
| Activation energy $Q$ | $0 \mathrm{Jmol}^{-1}$ |
| Stress exponent $n$ | $4$ |
| Prefactor $B$ | $1.23 \cdot 10^{-48} \mathrm{Pa}^{-n} \mathrm{s}^{-1}$ |
| Initial viscosity $\mu_{init}$ | $2 \cdot 10^{23}$Pas |
| **Mantle** | |
| Constant density $\rho$ | $3,150 \mathrm{kgm}^{-3}$ |
| Activation volume $V$ | $0 \mathrm{m}^3 \mathrm{mol}^{-1}$ |
| Activation energy $Q$ | $0 \mathrm{Jmol}^{-1}$ |
| Stress exponent $n$ | $1$ |
| Prefactor $B$ | $5.0 \cdot 10^{-22} \mathrm{Pa}^{-n} \mathrm{s}^{-1}$ |
| Initial viscosity $\mu_{init}$ | $10^{21}$Pas |





**Table 7.** 3D Subduction model parameters

| Parameter | Value [Unit] |
|---|---|
| Domain length | $4,000$km |
| Domain width | $800$km |
| Domain height | $660$km |
| Gravitational acceleration | $9.81\mathrm{ms}^{-2}$ |
| Stokes solver tolerance | $10^{-5}$ |
| **Model 1** | |
| Effective viscosity $\mu_{eff}^{vp}$ | Eq. (11) |
| Minimum viscosity $\mu_{\mathrm{min}}$ | $10^{19}$ Pas |
| Maximum viscosity $\mu_{\mathrm{max}}$ | $1.57 \cdot 10^{23}$Pas |
| Capped viscosity $\mu_{eff}$ | Eq. (13) |
| Local resolution | $80 \times 16 \times 16-$ |
| | $1280 \times 256 \times 256$ el. |
| Max. nr. of nonlinear iterations NI | 100 |
| Nr. of cores | 104 |
| DOF/core | $\sim 65,000$ |
| Wall time | 2.5weeks |
| Model run time | 40My |
| **Model 2** | |
| Reference viscosity $\mu_{\mathrm{ref}}$ | $10^{21}$Pas |
| Effective viscosity $\mu_{eff}^{vp}$ | Eq. (12) |
| Minimum viscosity $\mu_{\mathrm{min}}$ | $10^{19}$Pas |
| Maximum viscosity $\mu_{\mathrm{max}}$ | $10^{24}$Pas |
| Capped viscosity $\mu_{eff}$ | Eq. (14) |
| Thermal conductivity $k$ | $2.0\mathrm{Wm}^{-1}\mathrm{K}^{-1}$ |
| Thermal expansivity $\alpha$ | $2.0 \cdot 10^{-5}\mathrm{K}^{-1}$ |
| Reference temperature $T_0$ | 293K |
| Local resolution | $80 \times 16 \times 16-$ |
| | $640 \times 128 \times 128$ el. |
| Max. nr. of nonlinear iterations NI ($t_0$;rest) | 100;10 |
| Relative residual tolerance | $5.0 \cdot 10^{-5}$ |
| Nr. of cores | 260 |
| DOF/core | $\sim 132,500$ |
| Wall time | 6weeks |
| Model run time | 84My |



**Table 8.** 3D Subduction model parameters (based on Hirth and Kohlstedt (2003) and Ranalli (1995))

| Parameter | Mantle | OP | Lithosphere SP | Crust SP | AP | WZ | Unit |
|---|---|---|---|---|---|---|---|
| **Model 1** | | | | | | | |
| Activation volume $V_{dl}$ | 0.0 | 0.0 | 0.0 | 0.0 | | | $\mathrm{m^3mol^{-1}}$ |
| Activation energy $Q_{dl}$ | 0.0 | 0.0 | 0.0 | 0.0 | | | $\mathrm{Jmol^{-1}}$ |
| Stress exponent $n$ | 1.0 | 1.0 | 1.0 | 1.0 | | | - |
| Prefactor $B_{dl}$ | $2.12 \cdot 10^{-21}$ | $1.06 \cdot 10^{-23}$ | $3.03 \cdot 10^{-24}$ | $2.12 \cdot 10^{-24}$ | | | $\mathrm{Pa^{-n}s^{-1}}$ |
| Internal angle of friction $\phi$ | 0.0 | 0.0 | 0.0 | 0.0 | | | ° |
| Cohesion $C$ | $1.0 \cdot 10^{15}$ | $1.0 \cdot 10^{15}$ | $1.0 \cdot 10^{15}$ | $2.0 \cdot 10^{7}$ | | | Pa |
| Initial viscosity $\mu_{\mathrm{init}}$ | $1.57 \cdot 10^{20}$ | $3.14 \cdot 10^{22}$ | $4.71 \cdot 10^{22}$ | $1.57 \cdot 10^{23}$ | | | Pas |
| Constant density $\rho$ | $3,250$ | $3,250$ | $3,330$ | $3,330$ | | | $\mathrm{kgm^{-3}}$ |
| **Model 2** | | | | | | | |
| Activation volume $V_{df}$ | $5.0 \cdot 10^{-6}$ | $6.0 \cdot 10^{-6}$ | $6.0 \cdot 10^{-6}$ | 0.0 | 0.0 | 0.0 | $\mathrm{m^3mol^{-1}}$ |
| Activation energy $Q_{df}$ | $2.4 \cdot 10^{5}$ | $3.0 \cdot 10^{5}$ | $3.0 \cdot 10^{5}$ | 0.0 | 0.0 | 0.0 | $\mathrm{Jmol^{-1}}$ |
| Prefactor $B_{df}$ | $3.73 \cdot 10^{-14}$ | $6.08 \cdot 10^{-14}$ | $6.08 \cdot 10^{-14}$ | 0.0 | 0.0 | 0.0 | $\mathrm{Pa^{-n}s^{-1}}$ |
| Scaling factor $\beta_{df}$ | 0.5 | 1.0 | 1.0 | 2.0 | 2.0 | 2.0 | - |
| Activation volume $V_{dl}$ | $15 \cdot 10^{-6}$ | $20 \cdot 10^{-6}$ | $20 \cdot 10^{-6}$ | 0.0 | 0.0 | 0.0 | $\mathrm{m^3mol^{-1}}$ |
| Activation energy $Q_{dl}$ | $4.3 \cdot 10^{5}$ | $5.4 \cdot 10^{5}$ | $5.4 \cdot 10^{5}$ | 0.0 | 0.0 | 0.0 | $\mathrm{Jmol^{-1}}$ |
| Prefactor $B_{dl}$ | $3.91 \cdot 10^{-15}$ | $2.42 \cdot 10^{-16}$ | $2.42 \cdot 10^{-16}$ | $1.0 \cdot 10^{-19}$ | $1.0 \cdot 10^{-24}$ | $1.0 \cdot 10^{-21}$ | $\mathrm{Pa^{-n}s^{-1}}$ |
| Stress exponent n | 3.0 | 3.5 | 3.5 | 1.0 | 1.0 | 1.0 | - |
| Scaling factor $\beta_{dl}$ | 0.5 | 1.0 | 1.0 | 2.0 | 2.0 | 2.0 | - |
| Internal angle of friction $\phi$ | 20 | 0.0 | 20 | 0.0 | 0.0 | 0.0 | ° |
| Cohesion $C$ | $10^{6}$ | $10^{15}$ | $10^{6}$ | $10^{15}$ | $10^{15}$ | $10^{15}$ | Pa |
| Initial viscosity $\mu_{\mathrm{init}}$ | $2.0 \cdot 10^{20}$ | $5.4 \cdot 10^{23}$ | $5.4 \cdot 10^{23}$ | $1.0 \cdot 10^{20}$ | $5.4 \cdot 10^{23}$ | $1.0 \cdot 10^{21}$ | Pas |
| Specific heat $c_p$ | $1,250$ | $1,250$ | $1,250$ | 750 | $1,250$ | $1,250$ | $\mathrm{Jkg^{-1}K^{-1}}$ |
| Reference density $\rho_0$ | $3,350$ | $3,350$ | $3,350$ | $3,150$ | $3,350$ | $3,350$ | $\mathrm{kgm^{-3}}$ |

* For a fixed grain size of 0.01m (Hirth and Kohlstedt, 2003), length of Burgers vector of 0.5nm (Karato and Wu, 1993) and a shear modulus of 80GPa (Karato and Wu, 1993)





**Table 9.** The self-consistent subduction benchmark model parameters

| Parameter | Value [Unit] |
|---|---|
| Domain width | $3,000$km |
| Domain height | 750km |
| Local resolution | $256 \times 64$–$2,048 \times 512$ el. |
| Gravity acceleration | $9.81$ms$^{-2}$ |
| Stokes solver tolerance | $10^{-7}$ |
| Model end time | 100My |
| Temperature polynomial degree | 1 |
| Nr. of cores | 8–18 |
| Wall time | 27–71hr |
| **Sticky-air** | |
| Constant density $\rho$ | 1kgm$^{-3}$ |
| Constant viscosity $\mu$ | $10^{19}$Pas |
| **Subducting lithosphere** | |
| Constant density $\rho$ | $3,300$kgm$^{-3}$ |
| Constant viscosity $\mu$ | $10^{23}$Pas |
| **Mantle** | |
| Constant density $\rho$ | $3,200$kgm$^{-3}$ |
| Constant viscosity $\mu$ | $10^{21}$Pas |

**Table 10.** Wall time for timestep $2,000$ of the self-consistent subduction benchmark for different viscosity averaging methods using 28 cores

| Averaging method | Wall time $t_{2000}$ [s] |
|---|---|
| Arithmetic | $9.76 \cdot 10^4$ |
| Infinite norm | $3.84 \cdot 10^5$ |
| Geometric | $7.52 \cdot 10^4$ |
| Harmonic | $5.88 \cdot 10^4$ |

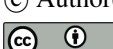



**Table 11.** Compressible subduction model parameters

| Parameter | Value [Unit] |
|---|---|
| Domain length | $1,600$km |
| Domain height | $1,000$km |
| Local resolution | $1.6 \times 1.6$– |
| | $25 \times 25$km |
| Gravitational acceleration | $9.81$ms$^{-2}$ |
| Surface temperature | $273$K |
| Mantle potential surface temperature | $1,600$K |
| Thermal conductivity $k$ | $4.0$Wm$^{-1}$K$^{-1}$ |
| Specific heat $c_p$ | $1,000$Jkg$^{-1}$K$^{-1}$ |
| Thermal expansivity $\alpha$ | $3.0 \cdot 10^{-5}$K$^{-1}$ |
| Reference viscosity $\mu_{\mathrm{ref}}$ | $10^{21}$Pas |
| Viscosity averaging | Eq. (16) |
| Effective viscosity $\mu_{eff}^{vp}$ | Eq. (12) |
| Minimum viscosity $\mu_{\mathrm{min}}$ | $10^{20}$Pas |
| Maximum viscosity $\mu_{\mathrm{max}}$ | $10^{24}$Pas |
| Viscosity capping $mu_{eff}$ | Eq. (13) |
| Compressibility $\beta$ | $5.124 \cdot 10^{-12}$Pa$^{-1}$ |
| 410 km Clapeyron slope $\gamma_{410}$ | $2.0 \cdot 10^{6}$PaK$^{-1}$ |
| 660 km Clapeyron slope $\gamma_{660}$ | $-1.5 \cdot 10^{6}$PaK$^{-1}$ |
| Transition widths $d$ | $10$km |
| 410 km density contrast $\delta_{\rho 410}$ | $273$kgm$^{-3}$ |
| 660 km density contrast $\delta_{\rho 660}$ | $342$kgm$^{-3}$ |
| 410 km transition pressure $P_{410}$ | $1.325 \cdot 10^{10}$Pa |
| 660 km transition pressure $P_{660}$ | $2.16 \cdot 10^{10}$Pa |
| Max. nr. of nonlinear iterations NI | $50$ |
| Surface pressure normalization | no |
| Nr. of cores | $90$ |
| DOF/core | $\sim 13,000$ |
| Wall time | $\sim 69$h |




**Table 12.** Compressible subduction material parameters

| Parameter | LM | TZ | UM | Crust SP | Crust OP | Unit |
|---|---|---|---|---|---|---|
| Activation volume $V_{df}$ | $1.0 \cdot 10^{-6}$ | $4.0 \cdot 10^{-6}$ | $4.0 \cdot 10^{-6}$ | $4.0 \cdot 10^{-6}$ | $4.0 \cdot 10^{-6}$ | $m^3 mol^{-1}$ |
| Activation energy $Q_{df}$ | $1.0 \cdot 10^5$ | $3.35 \cdot 10^5$ | $3.35 \cdot 10^5$ | $3.35 \cdot 10^5$ | $3.35 \cdot 10^5$ | $Jmol^{-1}$ |
| Prefactor $B_{df}$ | $1.0 \cdot 10^{-19}$ | $5.92 \cdot 10^{-11}$ | $5.92 \cdot 10^{-11}$ | $1.92 \cdot 10^{-11}$ | $1.92 \cdot 10^{-11}$ | $Pa^{-n}s^{-1}$ |
| Scaling factor $\beta_{df}$ | $1.0$ | $1.0$ | $1.0$ | $1.0$ | $1.0$ | - |
| Activation volume $V_{dl}$ | $14 \cdot 10^{-6}$ | $14 \cdot 10^{-6}$ | $14 \cdot 10^{-6}$ | $0.0$ | $0.0$ | $m^3 mol^{-1}$ |
| Activation energy $Q_{dl}$ | $4.0 \cdot 10^5$ | $4.0 \cdot 10^5$ | $4.0 \cdot 10^5$ | $0.0$ | $2.23 \cdot 10^5$ | $Jmol^{-1}$ |
| Prefactor $B_{dl}$ | $5.5 \cdot 10^{-20}$ | $5.5 \cdot 10^{-16}$ | $5.5 \cdot 10^{-16}$ | $1.0 \cdot 10^{-21}$ | $1.1 \cdot 10^{-28}$ | $Pa^{-n}s^{-1}$ |
| Stress exponent $n$ | $3.0$ | $3.0$ | $3.0$ | $1.0$ | $4.0$ | - |
| Scaling factor $\beta_{dl}$ | $2.0$ | $2.0$ | $2.0$ | $1.0$ | $1.0$ | - |
| Internal angle of friction $\phi$ | $30$ | $30$ | $30$ | $0.0$ | $0.0$ | $^\circ$ |
| Cohesion $C$ | $2.0 \cdot 10^7$ | $2.0 \cdot 10^7$ | $2.0 \cdot 10^7$ | $2.0 \cdot 10^7$ | $1.0 \cdot 10^8$ | Pa |
| Initial viscosity $\mu_{init}$ | $1.0 \cdot 10^{22}$ | $1.0 \cdot 10^{21}$ | $1.0 \cdot 10^{20}$ | $5.0 \cdot 10^{22}$ | $5.0 \cdot 10^{22}$ | Pas |
| Reference temperature $T_0$ | $1,800$ | $1,800$ | $1,800$ | $400$ | $400$ | K |
| Reference density $\rho_0$ | $3,915$ | $3,575$ | $3,300$ | $3,150$ | $2,900$ | $kgm^{-3}$ |



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
