# Peer review of "Implementing nonlinear viscoplasticity in ASPECT: benchmarking and applications to 3D subduction modeling"

_Solid Earth, 2017_

## Referee Comment (RC1) · Dr. May (Referee) · 17 Mar 2017

This paper discusses new functionality which has been incorporated into ASPECT (the advanced Solver for Problems in Earth's ConvecTion) to facilitate the simulation of 4D, regional scale subduction scenarios. The new functionality includes: (i) the definition of additional flow laws, and (ii) the introduction of a non-linear solver for the Stokes problem. The flow laws and plasticity models introduced into ASPECT, as well as the successive substitution strategy used to solve the non-linear Stokes problem are standard within the geodynamics community. That said, the intent of this submission is not to highlight new methodologies, but rather to highlight new functionality within a growing community code. Hence, the bulk of the paper is focused on demonstrating

the solution behaviour / correctness and performance one can expect when using this version of ASPECT to solve a variety of common geodynamic problems which involve non-linear flow laws.

I believe there is definitely merit in reporting on new functionality in community codes, and demonstrating how such functionality behaves on a standard set of relevant reference models. The latter point has been thoroughly addressed in this submission. I have some minor criticisms in regards to some of the methodological description provided. In some places it is unclear, or incomplete. These issues can all be easily addressed in the revised manuscript and I hope the comments I've made below will help with this clarification.

In my opinion, one of the best things about open-source projects is the transparency and reproducibility of the results. To encourage new users to exploit the new functionality described in this paper, and to enable them to reproduce your results (the former being a primary objective of this work), you have to provide precise information about the version of the software used, where the software can be obtained, and how to conduct the experiments you have presented. This has not been done to my satisfaction in this submission. As an excellent example of how this can be done, I refer you to a recent publication:

```
Wilson, Cian R. and Spiegelman, Marc and van Keken, Peter E.,
TerraFERMA: The Transparent Finite Element Rapid Model
Assembler for multiphysics problems in Earth sciences,
Geochemistry, Geophysics, Geosystems, 18 (2), 2016.
```

I encourage you to expand the supplementary material to include sufficient information required to reproduce your results. New users to ASPECT will greatly benefit from this addition.

Below I outline some general concerns, and following that I provide a number of de-

tailed corrections which should be addressed in the revised manuscript.

**General comments**

1. The title of the paper is not appropriate. The majority of the paper is focused on benchmarking / verification. There is very little content related to the actual implementation details. Please choose a more appropriate title which is consistent with the focal point of your paper.

2. Two of the stated objectives of the paper were (i) "provide hands-on examples" and (ii) to provide "community code for high-resolution, nonlinear rheology subduction modeling".

   To facilitate these points, for each reference model presented in this paper, you need to provide specific details defining: (a) where all necessary input files / data can be located (e.g. provide the a URL pointing us to your branch, pull request, web-page); (b) any special instructions required to run each reference model. Currently in Sec. 7, it just says "Input parameter files to reproduce the benchmarks will be incorporated as well." I don't know what this means. I scanned through the ASPECT GitHub repository and couldn't find the input files which define your models. Also, your ASPECT citation in the reference list says "developer version" - what does that mean? The master branch? Please clarify these points.

   Reproducibility of results from open-source codes should be possible. To facilitate this, you should provide the exact release / version number, or Git hash of ASPECT which was used for this study. Stating "The plasticity formulation has become part of the ASPECT distribution" is incomplete and does not enable an interested user to reproduce your results (assuming they had access to your input data).
[Figure]

3. When reporting the value and units of quantities, (i) please leave a single white space character between the value and the unit. When writing the unit, please leave a small half space between any two units. e.g. write Pa s and not Pas. Use the tex command \, for the half space.

4. Please punctuate all equations in the manuscript.

5. Throughout the paper, there are several instances where new features, or recently added features to ASPECT are mentioned. e.g. "(and, since recently, tracers)" and "Note that as of 2016 it is also possible to use active as well as passive tracers in ASPECT (version 1.4.0)." Your manuscript should concisely describe the method you used for the studies presented. Your discussion section should relate to the results you have presented. When you provide throughout the text, notes or remarks about features outside the scope of your results, you break the flow of the text.

   All material related to new features, or upcoming features should be confined to your "outlook" section. Please move all mention of tracers and Newton solvers into the outlook section as these components are not within the scope of this paper.

6. There are a number of missing details and undefined quantities in the methods section of the manuscript (Sec. 2) which are required to understand the exact implementation being used in ASPECT. These need to be addressed in the revision. I'll highlight the specific issues in the Corrections section below. To be a useful guide for users of ASPECT who wish to conduct experiments with non-linear flow laws, the underlying non-linear solver needs to be clearly defined.

7. I think there is little value in citing papers which are "in prep." as no one can access them, or read them (in whatever state they are in). As such, the citation is pointless. Please remove all citations to the "in prep." papers.

**Corrections**

1. [pg. 1, line 10] Your study doesn't involve "validation". This term is used to make a statement about whether the PDE you chose accurately describes a physical process (e.g. a lab experiment). Your study is concerned with "verification" which involves confirming that your implementation correctly solves the PDEs. Please change all instances of the word validate (and validation) to verify (verification).

2. [pg. 3, line 25] Please re-phase the sentence to be "Default settings employ second order polynomials for velocity and first order polynomials for pressure (Q2Q1 elements, e.g. Donea and Huerta, 2003), and second order polynomials for temperature and composition."

3. [pg. 4, line 5] The discrete form of equationss (3) and (4) will result in a non-symmetric operator. You cannot use the conjugate gradient method to solve this system. CG is for symmetric positive definite systems. Furthermore, the entropy viscosity method is by definition non-linear as the artificial viscosity is a function of the scalar (in your case $c_i$ or $T$). How are you solving this non-linear problem?

4. [pg. 4, line 5] Your statement about how you terminate the non-linear solver is incomplete. It should read something like this: "...until the relative nonlinear residual ... has fallen below a user-set tolerance (default value of $10^{-6}$), *or the user specified maximum number of iterations is reached.*"

5. [pg. 4, line 5] The variables $\mathbf{A}(\cdot)$, $\mathbf{b}$ and $\mathbf{x}$ have not been defined. Without this definition, I have no idea what your non-linear problem is, or how you are solving it. For example is $\mathbf{x} = (\mathbf{u}, \mathbf{p})$ or $(\mathbf{u}, \mathbf{p}, \mathbf{T})$? Each choice will change the definition of $\mathbf{A}(\cdot)$ and $\mathbf{b}$. I ask for clarification on this point as you solve an equation for $T$, and $T$ appears in your flow law.

[Figure]

6. [pg. 4, line 5] You state you use zero velocities to compute the initial residual. What value is used for the other quantities included in the definition of $\mathbf{x}$?

7. [pg. 4, line 5] You define the non-linear residual as $\mathbf{A}(\mathbf{x})\mathbf{x} - \mathbf{b}$. Defining it this way gives the reader the impression you might actually be computing the residual this way, e.g. by assembling a matrix and multiplying it by a vector. I hope that is not the case as this is an extremely inefficient way to evaluate the residual.

8. [pg. 4, line 15] Strain-rate is not a solution variable as you don't explicitly solve for $\epsilon_{ij}$. The strain-rate is a derived quantity obtained from the velocity solution variable.

9. [pg. 4, line 20] For rheology 1, why don't you just call it "Grain boundary sliding or diffusion creep".

10. [Eq. (14)] Suppose $\mu_{eff}^{vp} \approx 1$ throughout the domain, and I chose $\mu_{min} = 10^{-10}$ and $\mu_{max} = 10^{10}$. In this case, $\mu_{eff} \approx 1$ and this obviously causes no issues for the solver. Hence I think it is not meaningful to report you solved problems with $\mu_{max}/\mu_{min} = 10^7$ without specifying that the min/max limits were approached by the flow law adopted.

11. [pg. 5, line 10] If you examine Eq. 9, you'll notice that when $\phi = 0$, the expression you've written down does not reduce to the von Mises conditions (as you state it should). Please correct.

12. [pg. 5, line 25] "...avoid extreme excursion..." - what does this mean? Please re-phrase.

13. [pg. 5, line 25] Eq. (13) is stated in terms of $\eta$ whereas it should be stated in terms of $\mu$. Please correct.

14. [pg. 6, line 5] Regarding the sentence "...how to average their properties (viscosity, density and other)." Be specific and list all properties which are required to averaged. Don't say "other" as the reader has to guess what you actually mean. You never actually indicate how $\bar{\mu}$ is used in the finite element computations. If you replaced the symbol $\bar{\mu}$ with just $\mu$ there would be an obvious connection to Eq. 1. Furthermore, you should write or explain that $\mu_i$ is computed by evaluating Eq. 14 with the material constants for composition $i$.

15. [pg. 6, line 15] Please change "infinite norm" to "infinity norm". Please change all other instances of "infinite" to "infinity".

16. [Eq. (4)] When you introduce $c$, you should indicate that valid bounds of $c_i$. I think in your implementation you should enforce that $c_i \in [0, 1]$ but I have to guess that as it is not explained. Does the entropy viscosity actually enforce those bounds rigourously? I don't think your implementation introduces an limiters to enforce these bounds. What do you do in situations when $c_i < 0$ or $c_i > 1$? These details need to be explained somewhere in the manuscript.

17. [Eq. (5,6,8,9)] It would be useful if you defined these flow laws in a manner which made it clear which variables are constants associated with a particular composition ($i$); e.g.
$$\sigma_y = C_i \cos(\phi_i) + \sin(\phi_i)P,$$
where the index $i$ indicates a specific material (composition). I note you have done this (partially) in the tables of parameters, however I think adding an explicit sub-script $i$ on the constants in your flow law would be much clearer.

18. [Eq. (10)] You did not explicitly define what $\mu_{eff}^{df}$ and $\mu_{eff}^{dl}$ are.

19. [Eq. (18)] I don't understand your definition of the infinity norm as $\mu$ doesn't have

an index. I can think of two definitions:

$$\bar{\mu} = \max_{i=1,\dots,nc} \mu_i$$

or

$$\bar{\mu} = \mu_k,$$

where $k$ is compositional field index satisfying $c_k \geq c_i$ for all $i \neq k$. Please clarify your definition.

20. [pg. 6, line 25] The statement "All experiments were conducted on an in-house computer with 1, 000 cores" gives the reader the impression you conducted all experiments on 1000 cores, when you want to say that the machine you used has a 1000 cores. Please re-phase. Rather than tells as the clock speed (2.34 GHz), it would be more meaningful to report the type of compute node and the processor type.

21. [pg. 6, line 25] Remove the statement "Wall times quoted can have changed with versions of ASPECT newer than those used for the described experiments". Just provide information pertaining to your experiments - anything else is speculation. Your comment is vague and makes me think the run-times might have decreased with newer versions of ASPECT. In reality CPU times are impossible to reproduce anyway. Best thing is to report the machine spec, the compiler used (version) and leave it at that.

22. [Fig. 1] This figure is quite cluttered and unclear as you show the boundary conditions, the slip direction and try and label different regions within the solution. I suggest adding arrow heads to the red lines so the locations are more clearly defined.

23. [pg. 8, line 5] "...analytical solution is exactly reproduced ..." the numerical solution does not *exactly* match the analytic solution as you report 0.2% error. Re-phase.

24. [pg. 8, line 5] The statement "This trade-off is as expected, because the horizontal component of surface velocity is left free for the smooth punch, while it is fixed to zero for the rough punch" doesn't explain the discrepancy. Please remove this statement.

25. [pg. 8, line 15] Why are you taking about results related to 3D experiments when you models examine 2D solutions? Remove the following text as it's not relevant to your work or results. "In 3D, literature does suggest that a rough interface between indentor and medium results in a Prandtl slip-line geometry, while Hill?s solution is invoked by a smooth surface. Compare, for example, Fig. 11a and 11b of Gourvenec et al. (2006), Fig. 10e and 10f of Thieulot et al. (2008) or Fig. 13a and 13d of Braun et al. (2008)."

26. [Fig. 3] Please add to this figure snapshots of the pressure field.

27. [Fig. 3] Are you plotting a component of the strain-rate, or the second invariant? Please be more clear. The same comment applies for the velocity plot. Is this the magnitude of the velocity field?

28. [pg. 10, line 15] The statement "...The red symbols in Fig. 5 indicate runs for which the residual is not monotonously decreasing (after the first peak in residual)..." gives the impression you expect the residual to decrease monotonically. You use Picard without any type of globalization, so you are not guaranteed that the residuals will decrease monotonically.

29. [pg. 11, line 5] You have already justified why you consider pressure dependent plasticity models. I think you can remove (or relocate to your motivation sections) the sentence "As brittle failure in rocks is more appropriately described by pressure-dependent plasticity than by the perfectly-plastic deformation (Gerbault et al., 1998) used in the punch problem, our material model plugin includes frictional plasticity."

30. [pg. 14, line 10] "Through AMR, the total (velocity, pressure, temperature, composition) ..." these models don't include temperature so you should remove the word "temperature" from your statement.

31. [Fig. 10] Why is you adaptivity criterion performing so much refinement in the sticky-air? I can understand you want to resolve the air-rock interface, but refinement is occurring far from the interface. In one case, you have an isolated patch of refinement within the sticky air layer. Please comment on this.

32. [Fig. 10] In the caption you say "density leads to an elemental resolution varying from 512 x 128 to 32 x 8 elements". I presume this means an "effective" resolution, i,e. these are the element resolutions which correspond to the smallest and largest elements. I think it would be more clear if you just stated the min/max element edge length in the units used to define the model. This comment applies to all other descriptions of your results which involve an adaptive mesh.

33. [pg. 15, line 5] Regarding this statement: "Although the right shear band angles of 62 and 60 ..." Who is to say what the "right / corret" shear band angle is. Please re-phase.

34. [pg. 15, line 15] The following comment is incorrect "These are numerical effects tied to finite element models that should be taken into consideration when interpreting and comparing model results." What you are observing are not numerical effects. They are also not confined to finite element discretisations. The "effect" you are observing (lack of length scale) is due to the fact that your model configuration (specifically the geometry of you regions and boundary condition) creates singularities in the strain-rate field (and pressure field). With your plasticity formulation, this singularity wants to drive the shear band thickness to zero. However your numerical method cannot resolve the singularity, the best it can do is approximate it. This approximation improves as you refine the grid, and as a result your shear bands become thinner. We discuss this in Spiegelman et al "On the

solvability of incompressible Stokes with viscoplastic rheologies in geodynamics"
(2016).

35. [pg. 18, line 15] Again, it is not purely the rheology which is mesh dependent.
The lack of a length scale stems from your choice of geometry of the slab (sharp
corners) which induces singularities in the strain-rate field. If the problem is non-
linear, then the non-linear residual should always be monitored. There is no need
to make a special note of that here. Please remove the statement "...iterative
convergence should be monitored as for plastic rheologies."

36. [Fig. 18] Caption: Please clarify if you are plotting the strain-rate invariant.

37. [Fig. 18] Top panel. Please explain why the strain-rate (invariant?) field at the
upper surface (over riding plate side) contains discontinuities on the order of 1000
$s^{-1}$.

38. [pg. 24, line 20] Since the non-linear solver and rheology used by the ASPECT
models in Tosi et al differ from the implementation used in this work, you cannot
cite Tosi et al to support your verification study. Please remove the last part of
the first sentence in Sec. 5. Again, use the word verify and not validate.

39. [pg. 25, line 15] The term "Newton iterations" is inappropriate to describe the
methods used in Popov & Sobolev, May et al and Rudi et al. Newton is not an
iteration - it is a non-linear solver. Changing from Picard to Newton doesn't just
require change the iteration procedure. Many other solver components have to
be introduced. Please correct the text to reflect this. You should also add the
following paper to your list of geodynamics codes using Newton's method:

```
@article {GGGE:GGGE21224,
author = {Wilson, Cian R. and Spiegelman, Marc and
            van Keken, Peter E.},
title = {TerraFERMA: The Transparent Finite Element Rapid
```

```
          Model Assembler for multiphysics problems
          in Earth sciences},
journal = {Geochemistry, Geophysics, Geosystems},
volume = {18},
number = {2},
issn = {1525-2027},
url = {http://dx.doi.org/10.1002/2016GC006702},
doi = {10.1002/2016GC006702},
pages = {769--810},
year = {2017},
}
```

40. [Table 1] The parameter listed as "Effective deviatoric strain rate" is the symbol used to identify the second invariant of the strain-rate tensor. Please correct the parameter name so it is consistent with the rest of the text.

41. [Table 1] The symbol identified $\mu_{ref}$ identified with the name "Reference viscosity" does not appear in any equation shown in this paper. What is it? If it is not used - remove it from the table. It seems to appear in nearly every table, but I have no idea what this parameter actually means or how it relates to the rheological models used in this study.

42. [Table 5] As per an earlier comment, I think the parameter "Local resolution" would be better defined in terms of cm (in this model), rather than in terms of number of elements. Specifying the equivalent number of elements required if a structured, non-adaptive mesh was used is overly confusing.

43. [Table 6] The symbol for the reference viscosity given is $\mu_{max}$ - this looks like a typo.

44. [Table 6] The symbol for "viscosity capping" contains a latex typo.

45. [Table 8] The parameters $V_{dl}, Q_{dl}, B_{dl}$ have not been defined. These should be introduced when you define the specific flow laws for diffusion creep and dislocation creep.

---

## Referee Comment (RC2) · B.J.P. Kaus (Referee) · 12 May 2017

*Review of* **Implementing nonlinear viscoplasticity in ASPECT: benchmarking and applications to 3D subduction modeling** by Glerum, Thieulot, Fraters, Blom and Spakman

*Boris Kaus, Mainz*

This is a well-written manuscript in which the authors document their efforts to incorporated plasticity in the open-source ASPECT code, and demonstrate with a number of benchmarks, that the code gives similar results to that of other codes in our community. It also includes a new science application on 3D subduction with an adjacent plate and a few demonstrations of subduction models with more complex (nonlinear and compressible) rheologies. The paper is likely to be useful for others in the community and deserves to be published in SE.
Yet, I do have a few of minor to moderate issues, which I believe are important to address or at least discuss in more details before publication, as it will clarify the paper. All these can likely be incorporated without too much additional effort.

**1. Plasticity implementation**

The manuscript mainly deals with several benchmarks that demonstrate that the plasticity implementation gives similar results as that of other geodynamic codes. Yet, the implementation of plasticity itself is not described in very much detail, which is why the title of the paper is somewhat misleading (maybe drop "implementing" from the title. The plasticity implementation itself is also rather primitive (using only Picard iterations, for example), and some points deserve more discussion:
If you use a viscoplastic rheology, for example, you need an initial guess, which is usually done by taking a viscous-only step. You seem to use a user-specifiable constant viscosity for this, which can change from field to field. This can work (if you tune parameters just about right), but in my experience, viscoplastic models are extremely sensitive to this choice; if you use a too large value of viscosity, it won't converge to a physically sensitive solution because the initial guess for pressure (which is approximately equal to 4*EII*eta where EII is the strain rate and eta the viscosity) may be massively larger than what is physically meaningful. A background strain rate of 1e-15 1/s and initial viscosity of 1e24 Pas, for example, would result in a dynamic pressure of 4 GPa. Yet, in the setup of Figure 4, the lithostatic pressure at the base of the model should be on the order of 270 MPa, and for a frictional material (with friction angle 30 degree) the maximum pressure should be no more than about twice lithostatic (see Petrini & Podlachikov, 2001, among others). Thus, the dynamic pressure in this case is significantly larger than the physically admissible pressure, which can cause problems with convergence of the viscoplastic solution. So even while running the same model setup with ASPECT, different users could end up with totally different results simply because they had a different choice of initial viscosities. Since you implement this in a community code, it is important that you take care that users with less background in computational geodynamics don't produce physically meaningless results (I realize that this cannot be fully excluded, but you can at least try to minimize the chance for this to happen).

One way to do this different is to incrementally increase the boundary velocity (as described in Kaus 2010), during the first timestep. In practice, I found this is non-trivial to implement for more complicated model setups; moreover, it requires a large amount of initial iterations. A technique that is easier and more general (and which I have used since), is to

compute the admissible upper and lower bounds of pressure (that are usually a function of the lithostatic pressure), under the assumption of a homogeneous setup for a frictional material under compression & extension. The derivation of the admissible pressures is given in (Petrini & Podladchikov 2000) for a case with zero cohesion - you can easily extend that to a case with cohesion. During the first iteration step, or the first timestep, a pressure cutoff is applied within the yield function routine, which will ultimately limit the viscosities that the plasticity algorithm gives to reasonable. During subsequent iterations and subsequent timestep, this pressure cutoff is no longer applied. An advantage of this method is that it takes away 'user-tuning' of the initial viscosities. It is implemented in both MVEP2 and LaMEM (both available as open-source on bitbucket).

If you don't want to implement this in ASPECT, I would at least appreciate a longer discussion on the choice of reference viscosities and/or background strain rate and how new (or less experienced) users can detect non-sense results.

**2. Appendix A/B: Subduction benchmark:**

The benchmark setup you discuss has a 90 degree 'notch' and is the one which gave the worst results in the Schmeling benchmark. This may well be related to the 90 degree initial subduction angle which is very far removed from the angle the slab wants to make. A much better setup is case 3 considered in that paper, for which we also have laboratory experiments and for which the various codes had better agreement. It would be very interesting to see the effect of the viscosity averaging methods for this setup as well (I expect the discrepancies between the models to be much less severe). Can you add that?

**3. 3D Viscoplastic models**

Model setups 1 and 2 show that ASPECT can handle more complicated setups. Yet, from a science point of view, the initial geometry of the two cases is so different that it is difficult to discern what the effect of the adjacent plate is. You cite the paper of Schellart and Moresi (2013) in stating that the adjacent plate does not affect the geometry of the subducting plates. Yet on page 23 you also state that your rheology differs from their model. To make your conclusions more robust, it would be good if you can add at least one additional simulation with an adjacent plate for say the simple model setup (model 1). This would support your conclusions that the differences between models 1 and 2 are mainly caused by rheology and not by the adjacent plate.

Along similar lines, you mention that the plate viscosity of model 2 is an order of magnitude larger than that in model 1. From Fig. 19, it seems that the asthenosphere also has a lower viscosity, such that the overall slab-mantle viscosity contrast increases. I agree with you that a systematic study of these differences is probably beyond the current paper (even more since each 3D simulation takes several weeks). Yet, what would be interesting is to better understand whether the same effects are observed with 2D simulations as well, using exactly same setup but without adjacent plates. That should be computationally much faster and will at least give the reader some insights into how important 3D effects are for these kinds of subduction scenarios. It would be great

**4. Required wall-clock time and Picard vs. Newton iterations**

I really appreciate the honesty of the authors by reporting actual wall-clock times of the simulations, which gives interested readers a feeling for the computational costs involved in running ASPECT simulations. To be honest, the results left me a bit shocked. If a 3D free subduction simulation takes up to 6 weeks on 260 processors, with a maximum equivalent resolution of 640x128x128 elements, it essentially implies that it is nearly impossible to perform systematic science with this code (think about the carbon footprint

that this simulation has...). It is ofcourse difficult to make a one to one comparison to other codes (also since you did not report the number of timesteps in the tables - would be great if you can add this information). Yet, I'm routinely running 3D models with about the same maximum resolution (but without AMR) with LaMEM in a day or so on 64-128 cores. This thus at least suggests that there are massive differences between different codes and in terms of the science per CPU-hour such difference matter.

To get a somewhat better estimate of how timings differ and what the importance of Newton iterations is in this, I redid the 2D slab detachment benchmark (section 3.4) with LaMEM using your largest resolution (256x256 elements) on 8 cores. As LaMEM is currently a 3D-only code, the LaMEM simulation was done with 2 elements in the y-direction and employed 3 rather than 2 velocity components per node (such that the total degrees of freedom of the 256x256x2 simulations were with 590'848 slightly larger than the 456'400 DOFs used in the ASPECT simulation). In the LaMEM simulation, I reconstruct the slab thickness from the interpolated phase proportions on the staggered grid. This is likely to be slightly less precise than the marker-line approach used in your manuscript. Nevertheless, results are quite comparable to the Schmalholz solution:

[Figure]

Rather than taking 28 cores and 16 hours to compute the full simulation, the 295 timesteps of the LaMEM simulation were computed on 8 cores and took about 62 minutes. LaMEM thus seems to require (28*16*60)/(8*62)~55 times less CPU-time than ASPECT (assuming ASPECT employed a similar number of timesteps).

Part of this discrepancy may be caused by LaMEM employing a Newton nonlinear solver, rather than a Picard iterations. To understand how much this accounts for, I reran the simulation with Picard-only iterations and show the convergence behavior of timestep 10:

[Figure]

This demonstrates that Newton iterations are (in this case and for this timestep) around a factor 2 faster (note that we start every timestep with picard iterations before switching to the Newton solver). Evidently, for a tighter tolerance criteria the discrepancy between the two method is larger whereas it is less for a more relaxed nonlinear solver tolerance. Overall, the full simulation with Picard-only took around 153 minutes. So, whereas Newton can explain part of the discrepancy between the required wall-clock time for a full simulation, significant differences remain.

The input files for this setup, together with plotting routines, analysis tools and logfiles of the two simulations described above, are uploaded to the LaMEM repository under /input_models/DetachmentBenchmark.

It would be interesting to see how future optimizations of ASPECT (and of LaMEM) will reduce these timings and how this is in other geodynamic codes. The time-to-do-science is an important factor as well in computational geodynamics, that is unfortunately rarely documented for realistic cases (an exception being Pourhiet et al. 2017 for a different plasticity setup). I thus appreciate reporting these numbers – it would be great if you can report on how the latest ASPECT release affected the timings.

**Minor points:**
*p2. l14.*    It's LaMEM and the reference is wrong (should be Kaus et al., 2016)
*p3. l23.*    Please clarify whether you employed tracers here or not
*p4. l6/7:*   Please clarify whether these are the PETSc SNES options.
*p5. l7:*     Do you also use a zero initial guess for pressure or a lithostatic value?
*p5. eq(9):*  As far as I am aware most geodynamic codes employ the same yield-criteria in 2D as in 3D (so eq.8). That has the advantage that if you do pseudo-2D calculations with the 3D code (using say 1 element in the 3rd dimension) you

|  |  |
|---|---|
| | retrieve the 2D formulation. In your case, for typical values of the friction angle (30 degree), the 3D formulation deviates a few percent from the 2D one. |
| *p10. l2:* | Different than in Kaus (2010), you don't apply strain weakening in your setup. You do mention that later, but a comment at this stage would clarify things already. I would also appreciate a brief discussion on the choice of \mu_{init} on the model results. |
| *p12:* | You performed these simulations without AMR. What is the effect of using AMR on the shear band angles (if any)? |
| *p17: Fig 13:* | Did you mix the labelling of the x and y-axes of the figure? |
| *p17. l 4:* | The "first" three benchmarks (as the detachment benchmark is not plastic) |
| *p24, l21:* | You mention a benchmark of ASPECT that employs a different viscoplastic formulation. Can you explain better what the difference is? Do they not use a similar yield stress formulation and plastic viscosity? |
| *p25, l4/5:* | The first one to show the effect of a nonzero dilation angle in the geodynamics community was, as far as I am aware, the paper of Gerya & Yuen (2007) - see their figure 7. |
| *p25, l14/15:* | In my experience, adding elasticity significantly improves the convergence behavior of simulations with plastic failure (even though it does not solve all issues), and because of that it is worthwhile to incorporate. You are welcome to try MVEP2 or LaMEM to verify this. |
| *p25, l16/17:* | Newton iterations are crucial for fast convergence - you can add LaMEM and TerraFERMA to the list here. Yet, a pure viscoplastic rheology remains difficult to impossible to converge (as explained by Spiegelman et al., 2016). |
| *p26, sect. 6:* | Can you attach all scripts used to generate the benchmarks and figures to this paper, together with detailed instructions in the exact version of ASPECT you used to create the models? It seems likely that future code changes may give slighty different results; this way the interested reader has a reference point to reproduce your results. |
| *p27, l7:* | Why is the infinite norm computationally more expensive? Is that because you effectively end up with larger jumps in viscosity between adjacent elements, and you use iterative rather than direct solvers? |
| *p39, Table8:* | In model 1, I am a bit puzzled about the relationship between the B-parameter and the initial viscosity. These models are linearly viscous (apart from the crust), so why is \mu_{init} not simply 1/(2*B) as suggested by eq. 6? |

**Additional references:**

Petrini, K. & Podladchikov, Y., 2000. Lithospheric pressure-depth relationship in compressive regions of thickned crust. *Journal of Metamorphic Geology*, 18(1), pp.67–78.

Le Pourhiet L, May DA, Huille L, et al (2017) A genetic link between transform and hyper-extended margins. EPSL 465:184–192. doi: 10.1016/j.epsl.2017.02.043

---

## Author Comment (AC1) · 14 Jul 2017

**Response to Interactive comment of Dr. May (referee) on** "Implementing nonlinear viscoplasticity in ASPECT: benchmarking and applications to 3D subduction modeling" by Anne Glerum et al.

We thank Dr. May for his extensive and detailed comments, which greatly improved the manuscript. Below we address his points, and changes in the actual manuscript are indicated in bold.

General comments:

*1. The title of the paper is not appropriate. The majority of the paper is focused on benchmarking / verification. There is very little content related to the actual implementation details. Please choose a more appropriate title which is consistent with the focal point of your paper.*

We have changed the title to "Nonlinear viscoplasticity in ASPECT: benchmarking and applications to subduction".

*2. Two of the stated objectives of the paper were (i) "provide hands-on examples" and (ii) to provide "community code for high-resolution, nonlinear rheology subduction modeling".*
*To facilitate these points, for each reference model presented in this paper, you need to provide specific details defining: (a) where all necessary input files / data can be located (e.g. provide the a URL pointing us to your branch, pull request, web-page); (b) any special instructions required to run each reference model. Currently in Sec. 7, it just says "Input parameter files to reproduce the benchmarks will be incorporated as well." I don't know what this means. I scanned through the ASPECT GitHub repository and couldn't find the input files which define your models. Also, your ASPECT citation in the reference list says "developer version" - what does that mean? The master branch? Please clarify these points. Reproducibility of results from open-source codes should be possible. To facilitate this, you should provide the exact release / version number, or Git hash of ASPECT which was used for this study. Stating "The plasticity formulation has become part of the ASPECT distribution" is incomplete and does not enable an interested user to reproduce your results (assuming they had access to your input data).*

We have created a GitHub repository (https://github.com/anne-glerum/paper-aspect-plasticity-subduction-data) with all the input files and scripts to create them, plugins to ASPECT release 1.5 (https://github.com/geodynamics/aspect/tree/aspect-1.5) required for running the benchmarks, postprocessing scripts for all the gnuplot graphs and some of the ParaView images as well as instructions on how to use them.

We have changed the Code availability section to:

**Our simulations were performed with ASPECT version 1.5.0 (Bangerth et al. 2017), available on GitHub. This version includes the plastic rheology described in this paper as a material model plugin. It can also be found on https://github.com/anne-glerum/paper-aspect-plasticity-subduction-data, together with all the plugins and input files needed to reproduce the benchmarks and 3D subduction models. This directory includes postprocessing scripts to produce the plots in this paper as well.**

*3. When reporting the value and units of quantities, (i) please leave a single white space character between the value and the unit. When writing the unit, please leave a small half space between any two units. e.g. write Pa s and not Pas. Use the tex command \, for the half space.*
Done.

*4. Please punctuate all equations in the manuscript.*
Done.

*5. Throughout the paper, there are several instances where new features, or recently added features to ASPECT are mentioned. e.g. "(and, since recently, tracers)" and "Note that as of 2016 it is also possible to use active as well as passive tracers in ASPECT (version 1.4.0)." Your manuscript should concisely describe the method you used for the studies presented. Your discussion section should relate to the results you have presented. When you provide throughout the text, notes or remarks about features outside the scope of your results, you break the flow of the text.*
*All material related to new features, or upcoming features should be confined to your "outlook" section. Please move all mention of tracers and Newton solvers into the outlook section as these components are not within the scope of this paper.*
We have moved all mention of tracers, Newton solvers, material properties averaging (Appendix A3), strain tracking (Discussion) to the "Conclusions and Outlook" section.

*6. There are a number of missing details and undefined quantities in the methods section of the manuscript (Sec. 2) which are required to understand the exact implementation being used in ASPECT. These need to be addressed in the revision. I'll highlight the specific issues in the Corrections section below. To be a useful guide for users of ASPECT who wish to conduct experiments with non-linear flow laws, the underlying non-linear solver needs to be clearly defined.*
Done, see "Corrections" section below.

*7. I think there is little value in citing papers which are "in prep." as no one can access them, or read them (in whatever state they are in). As such, the citation is pointless. Please remove all citations to the "in prep." papers.*
Done.

Corrections:

1. *[pg. 1, line 10] Your study doesn't involve "validation". This term is used to make a statement about whether the PDE you chose accurately describes a physical process (e.g. a lab experiment). Your study is concerned with "verification" which involves confirming that your implementation correctly solves the PDEs. Please change all instances of the word validate (and validation) to verify (verification).*
Done.

2. *[pg. 3, line 25] Please re-phase the sentence to be "Default settings employ second order polynomials for velocity and first order polynomials for pressure (Q2Q1 elements, e.g. Donea and Huerta, 2003), and second order polynomials for temperature and composition."*
Done.

*3. [pg. 4, line 5] The discrete form of equations (3) and (4) will result in a nonsymmetric operator. You cannot use the conjugate gradient method to solve this system. CG is for symmetric positive definite systems. Furthermore, the entropy viscosity method is by definition non-linear as the artificial viscosity is a function of the scalar (in your case ci or T). How are you solving this non-linear problem?*
ASPECT's timestepping for the temperature equation has changed. Before, ASPECT used a semi-implicit BDF-2 scheme for the time discretization of this equation (see Kronbichler et al. 2012). The finite-difference approximation of the temperature (and velocity) time-derivatives at time $t^n$ was obtained through quadratic interpolation of temperature (and velocity) values at time $t^n$, $t^{n-1}$ and $t^{n-2}$. A linear interpolation of temperature $T^{n-1}$ and $T^{n-2}$ ($\mathbf{u}^{n-1}$ and $\mathbf{u}^{n-2}$) to find $T^n$ ($\mathbf{u}^n$) was then used in the advection term, while the diffusion term without the artificial diffusion was made implicit. Artificial diffusion was explicit by extrapolation. Hence the discrete form of the temperature/composition equations were spd and could be solved with CG, see also Eq. (16) of Kronbichler et al. (2012).

ASPECT 1.5 uses fully implicit time discretization. In this case, the system matrix is no longer spd and a GMRES solver with ILU preconditioner is used instead. Please refer to Heister et al. (2017) for a full discussion on this change in time discretization.

We have changed the sentence on the discretization of the temperature (and composition) equation to:

*The **GMRES** method with an incomplete LU decomposition preconditioner is used for the temperature and composition systems.*

*4. [pg. 4, line 5] Your statement about how you terminate the non-linear solver is incomplete. It should read something like this: "...until the relative nonlinear residual ... has fallen below a user-set tolerance (default value of 10  6), or the user specified maximum number of iterations is reached."*
We have changed the sentence to:

*... until the relative nonlinear residual ... has fallen below a user-set tolerance (default value of 1e-6)**, or the user-specified maximum number of iterations is reached**.*

*5. [pg. 4, line 5] The variables A(_), b and x have not been defined. Without this definition, I have no idea what your non-linear problem is, or how you are solving it. For example is x = (u; p) or (u; p;T)? Each choice will change the definition of A(_) and b. I ask for clarification on this point as you solve an equation for T, and T appears in your flow law.*
The solving of the temperature and Stokes equations is decoupled and the nonlinear iterations performed in this paper only concern the Stokes equations. Hence, x contains u and P only. Temperature and composition advection equations are solved once at the beginning of each time step. We have changed the description to the following:

Nonlinearities in the rheology are resolved with Picard- type (fixed point) iterations, iteratively updating the velocity and pressure, strain rate and viscosity (Ismail-Zadeh

and Tackley, 2010) until the relative nonlinear residual for iteration i $||A(x_{i-1})x_{i-1}-b||_2$ $/||A(x_0)x_0-b||_2$ has fallen below a user-set tolerance (default value of $10-6$**), or the user-specified maximum number of iterations is reached**. The initial residual $||A(x_0)x_0-b||_2$ is computed with zero velocities **and a lithostatic pressure profile calculated at the center horizontal coordinate. x contains the velocity and pressure solutions of the previous iteration, b represents the right hand side of the Stokes equations and A is the Stokes part of the system matrix.**

*[pg. 4, line 5] You state you use zero velocities to compute the initial residual. What value is used for the other quantities included in the definition of x?*
The initial guess for pressure considers a lithostatic pressure profile based on the model settings for density and gravity along the center of the domain. This pressure is also used in the computation of the initial residual.

*7. [pg. 4, line 5] You define the non-linear residual as A(x)x   b. Defining it this way gives the reader the impression you might actually be computing the residual this way, e.g. by assembling a matrix and multiplying it by a vector. I hope that is not the case as this is an extremely inefficient way to evaluate the residual.*
This is indeed how ASPECT computes the residual: the matrix is already assembled for the new solve and is then multiplied with the previous solution.

*8. [pg. 4, line 15] Strain-rate is not a solution variable as you don't explicitly solve for _ij . The strain-rate is a derived quantity obtained from the velocity solution variable.*
Done.

*9. [pg. 4, line 20] For rheology 1, why don't you just call it "Grain boundary sliding or diffusion creep".*
The atom migration either occurs along the grain boundaries or within the grains; we wished to express this explicitly.

*10. [Eq. (14)] Suppose mu_vp_eff = 1 throughout the domain, and I chose mu_min = 10e-10 and mu_max = 10e10. In this case, mu_vp_eff = 1 and this obviously causes no issues for the solver. Hence I think it is not meaningful to report you solved problems with mu_max/mu_min = 10e7 without specifying that the min/max limits were approached by the flow law adopted.*
Agreed. We meant that these limits were approached and have therefore rephrased the sentence to:

*We have successfully run the models presented here with **overall viscosity contrasts** of up to 7 orders of magnitude.*

*11. [pg. 5, line 10] If you examine Eq. 9, you'll notice that when _ = 0, the expression you've written down does not reduce to the von Mises conditions (as you state it should). Please correct.*
Done.

*12. [pg. 5, line 25] "...avoid extreme excursion..." - what does this mean? Please re-phrase.*
Rephrased to:

*... to avoid **extremely low or high viscosity values due to possible velocity anomalies feeding back into the rheology as well as large viscosity jumps** and thus ensure stability of the numerical scheme…*

*13. [pg. 5, line 25] Eq. (13) is stated in terms of eta whereas it should be stated in terms of mu. Please correct.*
Done.

*14. [pg. 6, line 5] Regarding the sentence "...how to average their properties (viscosity, density and other)." Be specific and list all properties which are required to averaged. Don't say "other" as the reader has to guess what you actually mean.*
Done.

*You never actually indicate how mu_average is used in the finite element computations. If you replaced the symbol mu_average with just mu there would be an obvious connection to Eq. 1.*
Done.

*Furthermore, you should write or explain that mu_i is computed by evaluating Eq. 14 with the material constants for composition i.*
We added to Section 2.2.2:

$\mu_i$ **is obtained by evaluating Eq. (11) or (12) using the material constants of composition i.**

*15. [pg. 6, line 15] Please change "infinite norm" to "infinity norm". Please change all other instances of "infinite" to "infinity".*
The term "infinite norm" was used by Schmeling et al. 2008, which we cite here. However, we have changed all instances to the generally accepted "infinity norm".

*16. [Eq. (4)] When you introduce c, you should indicate that valid bounds of ci. I think in your implementation you should enforce that ci [0; 1] but I have to guess that as it is not explained. Does the entropy viscosity actually enforce those bounds rigourously? I don't think your implementation introduces an limiters to enforce these bounds. What do you do in situations when ci < 0 or ci > 1? These details need to be explained somewhere in the manuscript.*
Initial conditions for the compositional field provide values on the [0,1] interval. Despite the entropy viscosity method, these limits can be slightly exceeded near the compositional boundaries; they are not enforced by the entropy viscosity method. Therefore, before using the field values for averaging, we cap them at 0 and 1. The division by the sum of the fields ensures proper averaging in case the fields do not add up to 1 in a particular point. We have added this explanation to the text:

**Note that each field $c_i$ is initialized with values on the interval [0,1] and capped values $0 \leq c_i \geq 1$ are used for averaging, as compositional field values may come to slightly exceed this interval over time despite artificial viscosity (Eq. (4)).**

*17. [Eq. (5,6,8,9)] It would be useful if you defined these flow laws in a mnner which made it clear which variables are constants associated with a particular composition (i); e.g._y = Ci cos(phi_i) + sin(phi_i)P;*
*where the index i indicates a specific material (composition). I note you have done this (partially) in the tables of parameters, however I think adding an explicit sub-script i on the constants in your flow law would be much clearer.*
We did not add the subscript i to the flow laws for two reasons: 1) We discuss the averaging *after* the flow laws are introduced. 2) We average the viscosities after computing the effective viscoplastic viscosity for each compositional field (i.e. after Eq. 12); which means the yield stresses and effective viscosities also require a subscript in Eq. 6-12, leading to very cluttered equations.

*18. [Eq. (10)] You did not explicitly define what mu_df eff and mu_dl eff are.*
We've changed the superscript of viscosity in Eq. (6) and added the following sentence after Eq. (6) to define them:

**The superscript df here indicates diffusion creep, dl dislocation creep.**

*19. [Eq. (18)] I don't understand your definition of the infinity norm as mu doesn't have an index. I can think of two definitions:*
*mu_av = max mu_i, i=1,...,nc*
*or*
*mu_av = mu_k;*
*where k is compositional field index satisfying c_k >= c_i for all i = k. Please clarify your definition.*
We intended the latter definition and have clarified it.

*20. [pg. 6, line 25] The statement "All experiments were conducted on an in-house computer with 1, 000 cores" gives the reader the impression you conducted all experiments on 1000 cores, when you want to say that the machine you used has a 1000 cores. Please re-phase. Rather than tells as the clock speed (2.34 GHz), it would be more meaningful to report the type of compute node and the processor type.*
Rephrased:

*All experiments were conducted on an in-house computer* **consisting of 1 Dell PE-R515 master node and 15 Dell PE-C6145 compute servers made up of 2x4 AMD Opteron 6136 CPUs with Qlogic InfiniBand QDR interconnect**. **ASPECT was compiled using GCC 4.9.2.**

*21. [pg. 6, line 25] Remove the statement "Wall times quoted can have changed with versions of ASPECT newer than those used for the described experiments". Just provide information pertaining to your experiments - anything else is speculation. Your comment is vague and makes me think the run-times might have decreased with newer versions of ASPECT. In reality CPU times are impossible to reproduce anyway. Best thing is to report the machine spec, the compiler used (version) and leave it at that.*
We have removed the phrase.

However, as our response to point 4 of reviewer Boris Kaus demonstrates, more recent versions of ASPECT do provide new ways of reducing run times. We have optimized wall times for the detachment benchmark discussed in point 4, but not for the other models. All benchmark run times are now reported for ASPECT 1.5.

For machine specs and compiler version, see point 20.

*22. [Fig. 1] This figure is quite cluttered and unclear as you show the boundary conditions, the slip direction and try and label different regions within the solution. I suggest adding arrow heads to the red lines so the locations are more clearly defined.*
We've tried to unclutter the figure by making a better distinction between boundary conditions and slip directions. Also we have removed the labels and used shaded areas instead. The footer of the figure now includes both velocity and pressure solutions for an unspecified punch velocity:

*Figure 1. Prandtl's analytical solution of a rigid die indenting a rigid-plastic half space (Davis and Selvadurai, 2002; Kachanov, 2004; Thieulot et al., 2008).* **Dark red arrows indicate the prescribed punch velocity $v_p$, shaded area CDE has a resulting velocity of $v_{CDE} = v_p$, while velocities in the lightest shaded areas are $v_{ABDC} = v_{EDFG} = v_p / \sqrt{2}$. Pressure at point I is $P_I = \sigma y(1+\pi)$ and $P_{ABC} = P_{EFG} = \sigma y$.**

*23. [pg. 8, line 5] "...analytical solution is exactly reproduced ..." the numerical solution does not exactly match the analytic solution as you report 0.2% error. Rephrase.*
Done:

*The analytical solution is* **reproduced with errors < 0.14% for a smooth punch**, *but the velocity vectors in Fig. 3g show some horizontal motion of triangle CDE and the velocity field is more diffuse.*

*24. [pg. 8, line 5] The statement "This trade-off is as expected, because the horizontal component of surface velocity is left free for the smooth punch, while it is fixed to zero for the rough punch" doesn't explain the discrepancy. Please remove this statement.*
Done.

*25. [pg. 8, line 15] Why are you taking about results related to 3D experiments when you models examine 2D solutions? Remove the following text as it's not relevant to your work or results. "In 3D, literature does suggest that a rough interface between indentor and medium results in a Prandtl slip-line geometry, while Hill?s solution is invoked by a smooth surface. Compare, for example, Fig. 11a and 11b of Gourvenec et al. (2006), Fig. 10e and 10f of Thieulot et al. (2008) or Fig. 13a and 13d of Braun et al. (2008)."*
Done.

*26. [Fig. 3] Please add to this figure snapshots of the pressure field.*
Done, see changed caption in point 27.

*27. [Fig. 3] Are you plotting a component of the strain-rate, or the second invariant? Please be more clear. The same comment applies for the velocity plot. Is this the magnitude of the velocity field?*
We are plotting the Frobenius norm of the strain rate and the magnitude of the velocity. We have changed the figure capture to reflect that:

*Figure 3. The punch benchmark results after 500 NI for a rough punch (left column) and a smooth punch (right column). (a) & **(f)**: Viscosity field with analytical slip lines. (b) & **(g)**: Strain rate **norm** ($\sqrt{\dot{\varepsilon} : \varepsilon}$) with measured shear band angles. (c) & **(h)**: Velocity **magnitude** with velocity vectors along the surface of the domain and velocity measurements in points K and L. (d) & **(i)**: **Pressure field**. (e) & (j) Pressure along the surface of the domain (colored line) and analytical solution values $\pi$ + 1 and 1 (grey lines). Rough punch: $P_I$ = 4.7382 and $P_H$ = $P_J$ = 0.6224. Smooth punch: $P_I$ = 4.1415 and $P_H$ = $P_J$ = 0.9999.*

*28. [pg. 10, line 15] The statement "...The red symbols in Fig. 5 indicate runs for which the residual is not monotonously decreasing (after the first peak in residual)..." gives the impression you expect the residual to decrease monotonically.*
*You use Picard without any type of globalization, so you are not guaranteed that the residuals will decrease monotonically.*
We rephrased as follows:

*The red symbols in Fig. 5 indicate runs for which the residual **did not drop below the convergence criterion $\varepsilon_u = 10^{-4}$ after 1000 iterations, as is evident from the corresponding red lines in Fig. 6.***

*29. [pg. 11, line 5] You have already justified why you consider pressure dependent plasticity models. I think you can remove (or relocate to your motivation sections) the sentence "As brittle failure in rocks is more appropriately described by pressure-dependent plasticity than by the perfectly-plastic deformation (Gerbault et al., 1998) used in the punch problem, our material model plugin includes frictional plasticity."*
This sentence is now the first sentence of Section 3.2.

*30. [pg. 14, line 10] "Through AMR, the total (velocity, pressure, temperature, composition)..." these models don't include temperature so you should remove the word "temperature" from your statement.*
Even though temperature is not considered in the setup of the experiment, the temperature system *is* set up for the nonlinear solver scheme that we picked. In runs with ASPECT 1.5, the solving of the temperature equation is skipped however.

*31. [Fig. 10] Why is you adaptivity criterion performing so much refinement in the sticky-air? I can understand you want to resolve the air-rock interface, but refinement is occurring far from the interface. In one case, you have an isolated patch of refinement within the sticky air layer. Please comment on this.*
The refinement was based on the norm of the strain rate and the approximate gradient of the density field (normalized to the same interval [0,1]) and a user-set percentage of the fraction of cells with the highest error that should be refined or coarsened. The refinement fraction was set to 95%, which led to some refinement in the sticky-air based on the strain rate there (see Fig. 9 b and f). We have rerun the sandbox with

refinement based on viscosity and density gradients, which greatly improves the focus of the refinement on the material interfaces and the shear bands. Figure 10 and it's caption are updated accordingly.

*32. [Fig. 10] In the caption you say "density leads to an elemental resolution varying from 512 x 128 to 32 x 8 elements". I presume this means an "effective" resolution, i,e. these are the element resolutions which correspond to the smallest and largest elements. I think it would be more clear if you just stated the min/max element edge length in the units used to define the model. This comment applies to all other descriptions of your results which involve an adaptive mesh.*
We have rephrased this particular caption as:

*Adaptive mesh refinement and coarsening based on the viscosity and density **leads to a minimum resolution of 6.25 × 6.25 mm and a maximum resolution of 0.39 × 0.39 mm.***

All other similar descriptions are also adapted.

*33. [pg. 15, line 5] Regarding this statement: "Although the right shear band angles of 62 and 60 ..." Who is to say what the "right / correct" shear band angle is. Please re-phase.*
We meant the shear bands directly to the right of the velocity discontinuity, not the 'correct' angle. Rephrased to:

*Although the shear band angles **to the right of the velocity discontinuity** of ...*

*34. [pg. 15, line 15] The following comment is incorrect "These are numerical effects tied to finite element models that should be taken into consideration when interpreting and comparing model results." What you are observing are not numerical effects. They are also not confined to finite element discretisations. The "effect" you are observing (lack of length scale) is due to the fact that your model configuration (specifically the geometry of you regions and boundary condition) creates singularities in the strain-rate field (and pressure field). With your plasticity formulation, this singularity wants to drive the shear band thickness to zero. However your numerical method cannot resolve the singularity, the best it can do is approximate it. This approximation improves as you refine the grid, and as a result your shear bands become thinner. We discuss this in Spiegelman et al "On the solvability of incompressible Stokes with viscoplastic rheologies in geodynamics" (2016).*
Agreed. We have changed the sentence to:

*As explained by Spiegelman et al. (2016), this lack of internal length scale is caused by the singularities in strain rate and pressure deriving from the model set-up (e.g. sharp corners of the silicon layer and the discontinuous velocity boundary condition) that are resolved better at higher resolutions, thereby decreasing shear band width.*

*35. [pg. 18, line 15] Again, it is not purely the rheology which is mesh dependent. The lack of a length scale stems from your choice of geometry of the slab (sharp corners) which induces singularities in the strain-rate field. If the problem is nonlinear, then the non-linear residual should always be monitored. There is no need to make a special note of that here. Please remove the statement "...iterative convergence should be monitored as for plastic rheologies."*
Yes, rephrased to:

*It should be noted **that the particular geometry of the slab with its sharp corners results in a mesh-dependence of the solution. Differences in model evolution can also arise from the particular viscosity and material averaging method applied.***

*36. [Fig. 18] Caption: Please clarify if you are plotting the strain-rate invariant.*
We plot the Frobenius norm of the strain rate and have added this description to all captions.

*37. [Fig. 18] Top panel. Please explain why the strain-rate (invariant?) field at the upper surface (over riding plate side) contains discontinuities on the order of 1000 s⁻1.*
For this model we applied an additional material averaging step where the viscosities and other material properties computed on the quadrature points of one element are averaged to obtain a constant value throughout the element. The contours of the plates cross different elements and therefore show the step-like discontinuities.

*38. [pg. 24, line 20] Since the non-linear solver and rheology used by the ASPECT models in Tosi et al differ from the implementation used in this work, you cannot cite Tosi et al to support your verification study. Please remove the last part of the first sentence in Sec. 5. Again, use the word verify and not validate.*
The non-linear solver used in the Tosi et al. paper was the same; the rheology was different. We have removed the last part of the sentence and used "verify".

*39. [pg. 25, line 15] The term "Newton iterations" is inappropriate to describe the methods used in Popov & Sobolev, May et al and Rudi et al. Newton is not an iteration - it is a non-linear solver. Changing from Picard to Newton doesn't just require change the iteration procedure. Many other solver components have to be introduced. Please correct the text to reflect this.*
We have changed the sentence to:

*The more **sophisticated and** efficient Newton **solver**…*

*You should also add the following paper to your list of geodynamics codes using Newton's method:*
*@article {GGGE:GGGE21224,*
*author = {Wilson, Cian R. and Spiegelman, Marc and*
*van Keken, Peter E.},*
*title = {TerraFERMA: The Transparent Finite Element Rapid Model Assembler for multiphysics problems*
*in Earth sciences},*
*journal = {Geochemistry, Geophysics, Geosystems},*

*volume = {18},*
*number = {2},*
*issn = {1525-2027},*
*url = {http://dx.doi.org/10.1002/2016GC006702},*
*doi = {10.1002/2016GC006702},*
*pages = {769--810},*
*year = {2017},}*
Done.

*40. [Table 1] The parameter listed as "Effective deviatoric strain rate" is the symbol used to identify the second invariant of the strain-rate tensor. Please correct the parameter name so it is consistent with the rest of the text.*
In Eq. (5), we define the symbol as the effective deviatoric strain rate, which is the square-root of the second moment invariant of the strain rate (Zienkiewicz and Taylor, 2002). We have added this definition to Table 1.

*41. [Table 1] The symbol identified mu_ref identified with the name "Reference viscosity" does not appear in any equation shown in this paper. What is it? If it is not used - remove it from the table. It seems to appear in nearly every table, but I have no idea what this parameter actually means or how it relates to the rheological models used in this study.*

The reference viscosity is used by ASPECT to compute a factor $\dfrac{\mu_{ref}}{L}$ for scaling the continuity equation (Eq. (2)) to obtain similar orders of magnitude for the dimensional momentum and mass equations (Eq. (1) and (2)). Characteristic length scale L should be set to a typical value for model features, while the choice of $\mu_{ref}$ should be guided by the viscosities present in the model. For variable viscosity models, this choice is not completely self-evident and as it affects the number of inner iterations, we listed the parameter value we used in the tables.

*42. [Table 5] As per an earlier comment, I think the parameter "Local resolution" would be better defined in terms of cm (in this model), rather than in terms of number of elements. Specifying the equivalent number of elements required if a structured, non-adaptive mesh was used is overly confusing.*
Changed to "Element size" for Table 5, 6, 7, 9 and 11.

*43. [Table 6] The symbol for the reference viscosity given is mu_max - this looks like a typo.*
Fixed.

*44. [Table 6] The symbol for "viscosity capping" contains a latex typo.*
Done.

*45. [Table 8] The parameters Vdl;Qdl;Bdl have not been defined. These should be introduced when you define the specific flow laws for diffusion creep and dislocation creep.*
They are declared in Table 1 and we've added the superscripts to Eq. 5 as well as the sentence after Eq. 5:
**The superscript df here indicates diffusion creep, dl dislocation creep.**

**Additional changes:**

The wall time for the indentor benchmark was quoted for the smooth indentor only, which was much smaller than for the rough indentor. We now report the wall time for both with ASPECT 1.5. Also, we changed the measurements of the velocity and pressure in Fig. 3, as it is now possible to extract solution variables at specific points based on the finite element solution instead of through ParaView.

The Stokes solver tolerance of the sandbox experiment was actually 1e-6 instead of the initially reported 1e-5.

**References**

Heister, T., Dannberg, J., Gassmöller, R., and Bangerth, W.: High accuracy mantle convection simulation through modern numerical methods
– II: realistic models and problems, Geophysical Journal International, 210, 833–851.

Kronbichler, M., Heister, T., and Bangerth, W.: High accuracy mantle convection simulation through modern numerical methods, Geophysical Journal International, 191, 12–29, 2012.

Schmeling, H. A. et al.: A benchmark comparison of spontaneous subduction models-Towards a free surface, Physics of the Earth and Planetary Interiors, 171, 198–223, 2008.

Tosi, N., Stein, C., Noack, L., Hüttig, C., Maierova, P., Samual, H., Davies, D. R., Wilson, C. R., Kramer, S. C., Thieulot, C., Glerum, A., Fraters, M., Spakman, W., Rozel, A., and Tackley, P. J.: A community benchmark for viscoplastic thermal convection in a 2-D square box, Geochemistry, Geophysics, Geosystems, 16, 2175–2196, 2015.

---

## Author Comment (AC2) · 14 Jul 2017

**Response to Interactive comment of Prof. Dr. Kaus (referee) on**
"Implementing nonlinear viscoplasticity in ASPECT: benchmarking and applications to 3D subduction modeling" by Anne Glerum et al.

We thank Prof. Dr. Kaus for his detailed review and reproducing one of the benchmarks, which allowed us to greatly improve the manuscript. Below we address his points, and changes in the actual manuscript are indicated in bold.

*General remarks:*
*1. Plasticity implementation*
*The manuscript mainly deals with several benchmarks that demonstrate that the plasticity implementation gives similar results as that of other geodynamic codes. Yet, the implementation of plasticity itself is not described in very much detail, which is why the title of the paper is somewhat misleading (maybe drop "implementing" from the title.*

We have dropped "implementing" from the title.

*The plasticity implementation itself is also rather primitive (using only Picard iterations, for example), and some points deserve more discussion: If you use a viscoplastic rheology, for example, you need an initial guess, which is usually done by taking a viscous-only step. You seem to use a user-specifiable constant viscosity for this, which can change from field to field. This can work (if you tune parameters just about right), but in my experience, viscoplastic models are extremely sensitive to this choice; if you use a too large value of viscosity, it won't converge to a physically sensitive solution because the initial guess for pressure (which is approximately equal to 4\*EII\*eta where EII is the strain rate and eta the viscosity) may be massively larger than what is*
*physically meaningful. A background strain rate of 1e-15 1/s and initial viscosity of 1e24 Pas, for example, would result in a dynamic pressure of 4 GPa. Yet, in the setup of Figure 4, the lithostatic pressure at the base of the model should be on the order of 270 MPa, and for a frictional material (with friction angle 30 degree) the maximum pressure should be no more than about twice lithostatic (see Petrini & Podlachikov, 2001, among others). Thus, the dynamic pressure in this case is significantly larger than the physically admissible pressure, which can cause problems with convergence of the viscoplastic solution. So even while running the same model setup with ASPECT, different users could end up with*
*totally different results simply because they had a different choice of initial viscosities. Since you implement this in a community code, it is important that you take care that users with less background in computational geodynamics don't produce physically meaningless results (I realize that this cannot be fully excluded, but you can at least try to minimize the chance for this to happen).*
*One way to do this different is to incrementally increase the boundary velocity (as described in Kaus 2010), during the first timestep. In practice, I found this is non-trivial to implement for more complicated model setups; moreover, it requires a large amount of initial iterations. A technique that is easier and more general (and which I have used since), is to compute the admissible upper and lower bounds of pressure (that are usually a function of the lithostatic pressure), under the assumption of a homogeneous setup for a frictional material under compression & extension. The derivation of the admissible pressures is*
*given in (Petrini & Podladchikov 2000) for a case with zero cohesion - you can easily extend that to a case with cohesion. During the first iteration step, or the first timestep, a pressure cutoff is applied within the yield function routine, which will ultimately limit the viscosities that the plasticity algorithm gives to reasonable. During subsequent iterations and subsequent timestep, this pressure cutoff is no longer applied. An advantage of this method is that it takes away 'user-tuning' of the initial viscosities. It is implemented in both MVEP2*

*and LaMEM (both available as open-source on bitbucket). If you don't want to implement this in ASPECT, I would at least appreciate a longer discussion on the choice of reference viscosities and/or background strain rate and how new (or less experienced) users can detect non-sense results.*

In ASPECT, during the first iteration step, we have the lithostatic pressure as initial pressure guess. Looking at your pressure cutoff in the yield function routine, this lithostatic pressure will lie within the pressure bounds calculated. Then prescribing the initial strain rate based on the velocity boundary conditions ensures the system starts in a reasonable state with a depth-dependent viscosity. The weak seed subsequently ensures localization of deformation and growing of the shear bands stemming from the seeds towards the surface.

If, on the other hand, we prescribe an initial high viscosity, the pressure resulting from the first nonlinear iteration might indeed be higher in magnitude than the lithostatic one. For example, for the 30 degrees compression case with an initial viscosity of 1e25 Pa s, the biggest pressure obtained after the first iteration is 452 MPa, which is still below the upper pressure limit which for this case would be 610 MPa. Even higher pressures would be 'corrected' by the viscosity cutoff, which will in fact set viscosity at the constant value $\mu_{max}$ (but not in the seed).

Figure 1 shows the shear bands resulting from 1000 nonlinear iterations but starting from a different initial viscosity for the medium. For low internal friction angles (in this case 10 degrees), there is no difference in the final shear band angle. However, for higher internal angles of friction (30 degrees), there is some variation in shear band angle for both compression and extension, even though the pressures obtained in the first iteration are within the pressure limits. This variation is at most 3 degrees and occurs for those runs only that are less well converged anyway (see Fig. 6 of the manuscript).

To the parameter table of each viscoplastic benchmark we have added a footnote on how an estimate of the initial viscosity or strain rate can be made.

[Figure]

**Figure 1 Shear band angles measured for different values of initial viscosity of the viscoplastic medium of the brick benchmark. Elemental resolution is 512x128. Two sets of models are performed in extension: one for an internal friction angle of 10 degrees and one of 30 degrees. In compression, a 30-degree friction angle is used.**

*2. Appendix A/B: Subduction benchmark:*
*The benchmark setup you discuss has a 90 degree 'notch' and is the one which gave the worst results in the Schmeling benchmark. This may well be related to the 90 degree initial subduction angle which is very far removed from the angle the slab wants to make. A much better setup is case 3 considered in that paper, for which we also have laboratory experiments and for which the various codes had better agreement. It would be very interesting to see the effect of the viscosity averaging methods for this setup as well (I expect the discrepancies between the models to be much less severe). Can you add that?*

To test the effect of the initial subduction angle, we have run a model based on case 3 of Schmeling et al. (2008) with ASPECT 1.5: we kept everything the same, except for the geometry of the slab tip (see adapted Figure 20 of the revised manuscript). The simulations indeed show much less divergence in the slab tip depth evolutions (see Figure 2 below that we also added to the paper). For example, instead of more than 95 My difference in minimum and maximum time for the slab tip to sink 300 km in case 1, the difference is only about 19 My. Trends in evolution with averaging method or mesh refinement level are still the same.

[Figure]

**Figure 2 Slab tip depth over time for case 3 of Schmeling et al. (2008) for four different averaging methods of the contribution of the compositional fields to viscosity. Colors indicate the averaging method, while one color goes from light to dark with local resolution, which varies from 256 × 64 elements to 2,048×512 elements. Minimum resolution is always 128×32 elements.**

*3. 3D Viscoplastic models*
*Model setups 1 and 2 show that ASPECT can handle more complicated setups. Yet, from a science point of view, the initial geometry of the two cases is so different that it is difficult to discern what the effect of the adjacent plate is. You cite the paper of Schellart and Moresi (2013) in stating that the adjacent plate does not affect the geometry of the subducting plates. Yet on page 23 you also state that your rheology differs from their model. To make your conclusions more robust, it would be good if you can add at least one additional simulation with an adjacent plate for say the simple model setup (model 1). This would support your conclusions that the differences between models 1 and 2 are mainly caused by rheology and not by the adjacent plate. Along similar lines, you mention that the plate viscosity of model 2 is an order of magnitude larger than that in model 1. From Fig. 19, it seems that the asthenosphere also has a lower viscosity, such that the overall slab-mantle viscosity contrast increases. I agree with you that a systematic study of these differences is probably beyond the current paper (even more since each 3D simulation takes several weeks). Yet, what would be interesting is to better understand whether the same effects are observed with 2D simulations as well, using exactly same setup but without adjacent plates. That should be computationally much faster and will at least give the reader some insights into how important 3D effects are for these kinds of subduction scenarios. It would be great*

We agree that the initial geometries of the two 3D cases are quite different. Therefore, we have run model 1 with a uniform adjacent plate (AP) and a transform zone (TZ) of the same uniform viscosity as the mantle until the slab tip is draping the bottom boundary of the domain. Both the AP and TZ are of the same thickness as the subducting plate. This model corresponds well with the findings of Schellart and Moresi (2013) in that the geometry of the subducting plate does not change in the presence of an adjacent plate, although in our case the subduction process is somewhat slowed down (TZ width is not given by Schellart and Moresi, 2013; another width could affect the subduction process differently.).

As the reviewer stated, a systematic study of the differences between model 1 and the thermo-mechanically coupled viscoplastic model 2 is beyond this paper. We have run a 2D simulation of model 2 that shows similar slab-mantle viscosity contrasts as the 3D model. Although subduction is faster for the 2D case than for the 3D case, this effect is also seen for model 1.

For more extensive investigations into rheological effects on subduction processes, see for example Andrews and Billen (2009) and Garel et al. (2014). Also, we invite the interested reader to investigate viscosity contrasts and other rheological controls on the subduction evolution with the input files of model 1 with the adjacent plate and the 2D thermo-mechanically coupled model that we have added to the repository belonging to this paper.

We have added the sentence to Section 4.3:
*A test with a uniform viscosity adjacent plate for our model 1 corroborates this.*

*4. Required wall-clock time and Picard vs. Newton iterations*
*I really appreciate the honesty of the authors by reporting actual wall-clock times of the simulations, which gives interested readers a feeling for the computational costs involved in running ASPECT simulations. To be honest, the results left me a bit shocked. If a 3D free subduction simulation takes up to 6 weeks on 260 processors, with a maximum equivalent resolution of 640x128x128 elements, it essentially implies that it is nearly impossible to perform systematic science with this code (think about the carbon footprint that this simulation has...). It is ofcourse difficult to make a one to one comparison to other codes (also since you did not report the number of timesteps in the tables - would be great if you can add this information). Yet, I'm routinely running 3D models with about the same maximum resolution (but without AMR) with LaMEM in a day or so on 64-128 cores. This thus at least suggests that there are massive differences between different codes and in terms of the science per CPU-hour such difference matter. To get a somewhat better estimate of how timings differ and what the importance of Newton iterations is in this, I redid the 2D slab detachment benchmark (section 3.4) with LaMEM using your largest resolution (256x256 elements) on 8 cores. As LaMEM is currently a 3D only code, the LaMEM simulation was done with 2 elements in the y-direction and employed 3 rather than 2 velocity components per node (such that the total degrees of freedom of the 256x256x2 simulations were with 590'848 slightly larger than the 456'400 DOFs used in the ASPECT simulation). In the LaMEM simulation, I reconstruct the slab thickness from the interpolated phase proportions on the staggered grid. This is likely to be slightly less precise than the marker-line approach used in your manuscript. Nevertheless, results are quite comparable to the*

*Schmalholz solution: --FIG--*

*Rather than taking 28 cores and 16 hours to compute the full simulation, the 295 timesteps of the LaMEM simulation were computed on 8 cores and took about 62 minutes. LaMEM thus seems to require (28\*16\*60)/(8\*62)~55 times less CPU-time than ASPECT (assuming ASPECT employed a similar number of timesteps).*

*Part of this discrepancy may be caused by LaMEM employing a Newton nonlinear solver, rather than a Picard iterations. To understand how much this accounts for, I reran the simulation with Picard-only iterations and show the convergence behavior of timestep 10: --FIG—*

*This demonstrates that Newton iterations are (in this case and for this timestep) around a factor 2 faster (note that we start every timestep with picard iterations before switching to the Newton solver). Evidently, for a tighter tolerance criteria the discrepancy between the two method is larger whereas it is less for a more relaxed nonlinear solver tolerance. Overall, the full simulation with Picard-only took around 153 minutes. So, whereas Newton can explain part of the discrepancy between the required wall-clock time for a full simulation, significant differences remain.*

*The input files for this setup, together with plotting routines, analysis tools and logfiles of the two simulations described above, are uploaded to the LaMEM repository under /input_models/DetachmentBenchmark.*

*It would be interesting to see how future optimizations of ASPECT (and of LaMEM) will reduce these timings and how this is in other geodynamic codes. The time-to-do-science is an important factor as well in computational geodynamics, that is unfortunately rarely documented for realistic cases (an exception being Pourhiet et al. 2017 for a different plasticity setup). I thus appreciate reporting these numbers – it would be great if you can report on how the latest ASPECT release affected the timings.*

Thank you for taking the time to run the detachment benchmark and provide us with wall time comparisons. The model run originally reported on in the paper was indeed performed with an older version of ASPECT (svn revision 1812) and underlying libraries. Also, time stepping settings were such that 1448 timesteps were taken. We have rerun the model with the latest ASPECT release (version 1.5, https://github.com/geodynamics/aspect/tree/aspect-1.5) and developer deal.II version (commit 1c58789f74fc4c7fd8ec82705ea24aeac8cedf84, https://github.com/dealii/dealii) and time stepping settings to match your 295 timesteps by 289 timesteps, leads to a wall time of 347 minutes on 28 cores. Using a cheaper Stokes solver (with a single V-cycle preconditioner) further reduces the wall time to 235 min (see Figure 3 for necking evolutions). Different compositional field averaging methods reduce wall times even further (this effect was previously shown in Table 10 of the paper), but fail to reproduce the necking curve of Schmalholz (2008). So does averaging of the material properties over each element (for a discussion on this averaging, see Heister et al. 2017). A run on the supercomputer Cartesius (bullx B720 nodes with 2x12 Intel Xeon E5-2690 v3 CPUs, Connect-IB InifiniBand) shows that the broader bandwidth and more modern hardware in general result in a wall time that is about 2.75 times faster. On 8 cores a run then takes 172 minutes, which is similar to the LaMEM wall time of 153 minutes.

[Figure]

**Figure 3** Necking evolution for different model parameters. Note that results for the same number of timesteps are identical.

We now report the necking evolution and wall time for the cheap Stokes solver run on our local cluster (235 min). Note that we have not performed wall time optimization for the other run times reported. Also we've added a section to the overall discussion elaborating on the different parameters affecting and possibly reducing wall time.

To Section 2.1.2 we have added:
*A cheap Stokes solve option is available in which the preconditioner employs one V-cycle only. The number of such FGMRES iterations before switching to the more expensive preconditioner is set to 0 in this paper, unless stated otherwise.*

To the results of the indentor benchmark (Section 3.1.2) we added:
**When using 200 cheap Stokes iterations for the smooth punch, results are not changed, but wall time is about 1.6 times longer. Using harmonic averaging of the material properties as discussed in Heister et al. (2017) increases the velocity error for the smooth punch to ~1%, but reduces wall time about 4.7 times. Loosening the linear Stokes solver tolerance by 1 order of magnitude to $10^{-8}$ reduces the wall time of the rough punch by a factor of 1.6, while keeping the velocity error < 1%.**

To the overall discussion we added:

*Nonlinear rheologies also affect the linear solver by introducing large viscosity gradients. Different strategies to reduce the increased computational time and under/overshooting of the numerical approximation of the viscosity/pressure gradient are available in ASPECT. For one, one can reduce the linear tolerance (while making sure the results do not change significantly), as was shown in Section 3.1. Secondly, a cheap Stokes solver can be employed, although this does not help for each model set-up (compare Sections 3.1 and 3.4). Thirdly, averaging the contributions of the compositional field to the viscosity and other material properties in a*

*specific point reduces the sharpness of viscosity boundaries, making the problem easier to solve, but with the choice of averaging method affecting the model evolution (Section3.4 30 and Appendix tab:schmelingmodel). Lastly, averaging of material properties such as viscosity and density over each element reduces pressure oscillations (Heister et al., 2017), but can also influence the model evolution as was shown in Section 3.4.*

Minor points:

*p2. l14. It's LaMEM and the reference is wrong (should be Kaus et al., 2016)*
Our apologies, we have changed the capitalization and the bibtex category.

*p3. l23. Please clarify whether you employed tracers here or not*
We have removed this sentence as we did not employ tracers except for the tracking of the necking evolution of the detachment benchmark.

*p4. l6/7: Please clarify whether these are the PETSc SNES options.*
They are not, we used ASPECT with Trilinos and not PETSc.

*p5. l7: Do you also use a zero initial guess for pressure or a lithostatic value?*
The initial guess for pressure considers a lithostatic pressure profile based on the model settings for density and gravity along the center of the domain. This pressure is also used in the computation of the initial residual.

*p5. eq(9): As far as I am aware most geodynamic codes employ the same yield-criteria in 2D as in 3D (so eq.8). That has the advantage that if you do pseudo-2D calculations with the 3D code (using say 1 element in the 3rd dimension) you retrieve the 2D formulation. In your case, for typical values of the friction angle (30 degree), the 3D formulation deviates a few percent from the 2D one.*
The 3D formulation is a text book Drucker--Prager formulation (e.g. Davis and Selvadurai 2002), a circumscribing cone of the hexagonal cone of the Mohr--Coulomb yield surface. In 2D incompressible plane strain, the Mohr--Coulomb and Drucker--Prager yield surfaces are identical. In case a pseudo--2D model with the same formulation as 2D is required, the user can easily select the formulation in Eq. (8) manually in the material model plugin.

*p10. l2: Different than in Kaus (2010), you don't apply strain weakening in your setup. You do mention that later, but a comment at this stage would clarify things already.*
We added the following sentence to Section 3.2.1:

**Strain softening of the cohesion and angle of internal friction of the medium is not incorporated.**

*p10. l2 I would also appreciate a brief discussion on the choice of $\mu_{init}$ on the model results.*
See also our reply to your general remark 1. To the manuscript we added the following in Section 3.2.2:

**Varying the initial viscosity of the viscoplastic medium from $\mu_{min}$ to $\mu_{max}$ for a uniform mesh of 512 × 128 elements leads to the same shear band angles for well-behaved residual runs (see black lines in Fig. 6), while for higher internal angle of friction runs, a variation of maximally 3◦ is found.**

*p12: You performed these simulations without AMR. What is the effect of using AMR on the shear band angles (if any)?*

We have performed several sets of runs both in compression and extension using AMR. Instead of doing 1000 nonlinear iterations on one mesh refinement level, we did 333 iterations on 3 different levels, with the finest level being of the same order as the fixed mesh refinement level. After 333 iterations, mesh refinement is performed based on the norm of the strain rate, viscosity or velocity. Differences in final shear band angles arise from the different areas that are refined for the different strategies and the user-set fractions of cells that should be refined. For uniform meshes, variation in angles with mesh size is already seen, and this variation can be carried on to finer meshes with AMR. The differences in shear band angle are shown in Figure 4: differences can be up to 5 degrees, but the angles remain within the theoretical Arthur--Coulomb range.

We've added a sentence to the results section of the brick benchmark (Section 3.2.2):

***To estimate the effect of adaptive mesh refinement on the shear band angles, we ran additional tests with 3 × 333 245 nonlinear iterations at increasing refinement levels, with refinement based on gradients in the velocity, viscosity or strain rate and different fractions of cells that are refined. These simulations indicate a maximal variation of 5∘ degrees compared to results for a uniform mesh of 512 × 128 elements.***

[Figure]

**Figure 4 Shear band angles for different internal angles of friction. Round symbols indicate runs with a uniform mesh resolution of 512x128 elements, while triangular symbols represent runs with a base mesh resolution of 128x32 elements and two levels of adaptive mesh refinement, amounting to the same local mesh resolution.**

*p17: Fig 13: Did you mix the labelling of the x and y-axes of the figure?*
Yes. This has been corrected.

*p17. l 4: The "first" three benchmarks (as the detachment benchmark is not plastic)*
Done.

*p24, l21: You mention a benchmark of ASPECT that employs a different viscoplastic formulation. Can you explain better what the difference is? Do they not use a similar yield stress formulation and plastic viscosity?*
In response to the first reviewer, we no longer mention this benchmark in the discussion. The viscoplastic rheology used by Tosi et al. (2015) consists of a harmonic average of a temperature- and depth-dependent linearized Arrhenius law and a nonlinear part that is the sum of a constant effective viscosity and a plastic

viscosity calculated as $\dfrac{\sigma_y}{\sqrt{\dot{\varepsilon}_{ij}\dot{\varepsilon}_{ij}}}$ .

*p25, l4/5: The first one to show the effect of a nonzero dilation angle in the geodynamics community was, as far as I am aware, the paper of Gerya & Yuen (2007) - see their figure 7.*
We added a reference to their paper:

*For example**, Gerya and Yuen (2007) included dilatant materials and** Choi and Petersen (2015) argue that numerical models should incorporate an initially associated plastic flow rule that evolves into a non-associated flow rule with increased slip to assure persistent Coulomb shear band angles while avoiding unlimited dilatation.*

*p25, l14/15: In my experience, adding elasticity significantly improves the convergence behavior of simulations with plastic failure (even though it does not solve all issues), and because of that it is worthwhile to incorporate. You are welcome to try MVEP2 or LaMEM to verify this.*
See the next point.

*p25, l16/17: Newton iterations are crucial for fast convergence - you can add LaMEM and TerraFERMA to the list here. Yet, a pure viscoplastic rheology remains difficult to impossible to converge (as explained by Spiegelman et al., 2016).*
We have added LaMEM and TerraFERMA to the list and rephrased the paragraph as follows:

*Incorporating more realistic nonlinear rheologies such as described in this paper creates the necessity for additional nonlinear iterations within a single time step. Also, we have seen that at higher mesh resolutions, more of such iterations are required 15 to converge the solution. This greatly increases model run time and therefore it is important to implement a more efficient nonlinear solving strategy than the Picard iterations currently used by ASPECT. The more **sophisticated** Newton **solver** (see for example Popov and Sobolev, 2008; May et al., 2015; Rudi et al., 2015; Kaus et al., 2016; Wilson et al. 2017) **will help achieving faster convergence.***

***Convergence behavior has also been suggested to improve from including elasticity (Kaus 2010), but especially dynamic pressure dependent plasticity remains difficult to converge for both Picard iterations and Newton solvers (Spiegelman et al. 2016).***

*p26, sect. 6: Can you attach all scripts used to generate the benchmarks and figures to this paper, together with detailed instructions in the exact version of ASPECT you used to create the models? It seems likely that future code changes may give slighty different results; this way the interested reader has a reference point to reproduce your results.*

We have put all the input files, scripts to generate them and the necessary plugins to ASPECT1.5 in a GitHub reposity (https://github.com/anne-glerum/paper-aspect-plasticity-subduction-data). This repository also includes the postprocessing and plotting scripts for the graphs.

*p27, l7: Why is the infinite norm computationally more expensive? Is that because you effectively end up with larger jumps in viscosity between adjacent elements, and you use iterative rather than direct solvers?*

Yes, the maximum norm prohibits any smoothing due to the gradual transition from one composition to the other, leading to larger viscosity jumps between quadrature points, which is harder on the solver.

*p39, Table8: In model 1, I am a bit puzzled about the relationship between the B-parameter and the initial viscosity. These models are linearly viscous (apart from the crust), so why is $\mu_{init}$ not simply 1/(2\*B) as suggested by eq. 6?*

The prefactor was calculated to mitigate a correction for uniaxiality for pure shear measurements of $\frac{1}{2}3^{\frac{n+1}{2}}B$ that our material model can take into account. However, this correction was not applied and the intended constant viscosities are off by a factor of 1.5 from the initial viscosities.

**Additional changes:**

The wall time for the indentor benchmark was quoted for the smooth indentor only, which was much smaller than for the rough indentor. We now report the wall time for both with ASPECT 1.5. Also, we changed the measurements of the velocity and pressure in Fig. 3, as it is now possible to extract solution variables at specific points based on the finite element solution instead of through ParaView.

The Stokes solver tolerance of the sandbox experiment was actually 1e-6 instead of the initially reported 1e-5.

**References**

Andrews, E. R. and Billen, M. I.: Rheologic controls on the dynamics of slab detachment, Tectonophysics, 2009.

Garel, F. et al.: Interaction of subducted slabs with the mantle transition-zone: a regime diagram from 2-D thermo-mechanical models with a mobile trench and an overriding plate, Geochemistry, Geophysics, Geosystems, 2014.

Heister, T., Dannberg, J., Gassmöller, R., and Bangerth, W.: High accuracy mantle convection simulation through modern numerical methods
– II: realistic models and problems, Geophysical Journal International, 210, 833–851.

Kronbichler, M., Heister, T., and Bangerth, W.: High accuracy mantle convection simulation through modern numerical methods, Geophysical Journal International, 191, 12–29, 2012.

Schellart, W. P. and Moresi, L.: A new driving mechanism for backarc extension and backarc shortening through slab sinking induced toroidal and poloidal mantle flow: Results from dynamic subduction models with an overriding plate, Journal of Geophysical Research, 118, 1–28, 2013.

Schmalholz, S. M.: A simple analytical solution for slab detachment, Earth and Planetary Science Letters, 304, 45–54, 2011.

Schmeling, H. A. et al.: A benchmark comparison of spontaneous subduction models-Towards a free surface, Physics of the Earth and Planetary Interiors, 171, 198–223, 2008.

Tosi, N., Stein, C., Noack, L., Hüttig, C., Maierova, P., Samual, H., Davies, D. R., Wilson, C. R., Kramer, S. C., Thieulot, C., Glerum, A., Fraters, M., Spakman, W., Rozel, A., and Tackley, P. J.: A community benchmark for viscoplastic thermal convection in a 2-D square box, Geochemistry, Geophysics, Geosystems, 16, 2175–2196, 2015.